# Antisense oligonucleotide-mediated *Dnm2* knockdown prevents and reverts myotubular myopathy in mice

Hichem Tasfaout[1,2,3,4], Suzie Buono[1,2,3,4], Shuling Guo[5], Christine Kretz[1,2,3,4], Nadia Messaddeq[2,3,6], Sheri Booten[5], Sarah Greenlee[5], Brett P. Monia[5], Belinda S. Cowling[1,2,3,4,*] & Jocelyn Laporte[1,2,3,4,*]

Centronuclear myopathies (CNM) are non-dystrophic muscle diseases for which no effective therapy is currently available. The most severe form, X-linked CNM, is caused by myotubularin 1 (*MTM1*) loss-of-function mutations, while the main autosomal dominant form is due to dynamin2 (*DNM2*) mutations. We previously showed that genetic reduction of DNM2 expression in *Mtm1* knockout (Mtm1KO) mice prevents development of muscle pathology. Here we show that systemic delivery of *Dnm2* antisense oligonucleotides (ASOs) into Mtm1KO mice efficiently reduces DNM2 protein level in muscle and prevents the myopathy from developing. Moreover, systemic ASO injection into severely affected mice leads to reversal of muscle pathology within 2 weeks. Thus, ASO-mediated DNM2 knockdown can efficiently correct muscle defects due to loss of MTM1, providing an attractive therapeutic strategy for this disease.

[1] Department of Translational Medicine and Neurogenetics, Institut de Génétique et de Biologie Moléculaire et Cellulaire (IGBMC), Illkirch 67404, France. [2] INSERM U964, Illkirch 67404, France. [3] CNRS UMR7104, Illkirch 67404, France. [4] FMTS, Strasbourg University, Illkirch 67404, France. [5] Ionis Pharmaceuticals Inc., Carlsbad, California 92010, USA. [6] Service de Microscopie Electronique, Institut de Génétique et de Biologie Moléculaire et Cellulaire (IGBMC), Illkirch 67404, France. * These authors contributed equally to this work. Correspondence and requests for materials should be addressed to B.S.C. (email: belinda@igbmc.fr) or to J.L. (email: jocelyn@igbmc.fr).

**Figure 1 | *In vitro* validation of ASOs targeting *Dnm2*.** (**a**) Location of ASO-targeting sequences in *Dnm2* mouse pre-mRNA. (**b**) *Dnm2* mRNA level in ASO-treated C2C12 myoblast cells was determined by qRT-PCR and standardized to *Hprt*. Cells were electroporated with ASO control (ctrl), #1, #2 or #3 at 0.015, 0.06, 0.25 or 1 μM and collected 24 h later. (**c**) Representative western blot from the same C2C12 cells for DNM2 and the GAPDH loading control. Below, DNM2 protein levels was determined by densitometry and standardized to GAPDH. n = 3 per each group. Data represent an average of three independent experiments ± s.e.m. *P < 0.05, ***P < 0.001 for ASO *Dnm2*-treated versus ASO ctrl-treated cells (ANOVA test). kDa, kilodalton; MW, molecular weight.

Antisense oligonucleotide (ASO) technology represents a promising therapeutic approach for neuromuscular diseases caused by gain-of-function mutations by reducing expression of the mutant gene, or to induce production of a functional or partially functional protein through a number of mechanisms including splicing modulation for loss-of-function genetic diseases[1–4]. To date, this approach has been mainly applied to dystrophic muscles where uptake may be facilitated by muscle fibre leakiness, or to diseases with nuclear accumulation of pathogenic RNA that favours RNase H1-dependent degradation of the targeted transcript[3]. X-linked centronuclear myopathy (XLCNM), or myotubular myopathy, is a non-dystrophic muscle disease characterized by muscle weakness and hypotrophic fibres with centralized nuclei[5–7]. It is due to loss-of-function mutations in the phosphoinositides phosphatase myotubularin (*MTM1*)[8]. Both patient muscle biopsies and the *Mtm1* knockout (Mtm1KO) mouse model present an overexpression of dynamin 2 (DNM2) as a consequence of *MTM1* mutations[9]. Of note, overexpression of DNM2 in wild-type (WT) mice by transgenesis or adeno-associated virus creates a centronuclear myopathy (CNM) phenotype[10,11], while genetic reduction of DNM2 in the myopathic Mtm1KO mice, by crossing with *Dnm2+/−* heterozygous mice, prevented the development of myotubular myopathy in Mtm1KO mice[9], altogether pointing to DNM2 as a target for therapeutic development. In addition, autosomal dominant CNM forms are due to heterozygous mutations in *DNM2* that are thought to increase the GTPase activity and oligomerization of DNM2 (refs 12–14).

To develop a therapeutic approach for XLCNM, we tested ASOs against *Dnm2* that act through a RNase H-dependent RNA degradation mechanism utilizing the recently developed constrained ethyl ASO chemistry (cEt)[15]. This ASO chemistry exhibits enhanced *in vitro* and *in vivo* potency through higher binding affinity and better protection against nuclease degradation when compared to other chemistries, while maintaining a favourable safety profile[16–18].

Here, we demonstrate that intramuscular injection and systemic delivery of *Dnm2* ASOs efficiently reduced DNM2 protein levels and rescue different CNM features in the Mtm1KO murine model of myotubular myopathy. These data confirm the epistasis between *Mtm1* and *Dnm2* and validate DNM2 knockdown as a therapeutic approach for XLCNM.

## Results

**In vitro and in vivo ASO screening for DNM2 downregulation.** Following a screen of ~500 ASOs, we identified three ASO candidates based on relative potency for reducing DNM2 levels (Fig. 1a, Supplementary Table 1), each of which displayed a strong dose-dependent knockdown of the *Dnm2* mRNA transcript (Fig. 1b) and protein (Fig. 1c) when electroporated into C2C12 mouse myoblasts. ASO#1 targets exon17, while ASO#2 and #3 target repeated sequences in intron 16 of the *Dnm2* pre-mRNA.

To assess the *Dnm2* knockdown efficacy of selected ASOs *in vivo* and to potentially rescue the myotubular myopathy phenotype, 20 μg of ASO#1, #2 or #3 were injected weekly into the right tibialis anterior (TA) of 3-week-old WT or Mtm1KO mice and the same dose of control ASO was injected into contralateral limbs. The Mtm1KO mouse develops a progressive muscle weakness starting from week 2 to 3 (ref. 19). They display most features observed in patients such as a decrease in muscle weight and force, ptosis, kyphosis and subsequent breathing difficulties that cause death by 8–9 weeks. Their muscles display typical CNM histology with predominance of small rounded fibres with mislocalized nuclei, abnormal mitochondria distribution and

alteration in triad shape and orientation[19–22]. In addition, their muscles display mislocalization of dihydropyridine receptor (DHPR) and Ryanodine receptor 1 (RYR1), two calcium channels that play an important role in excitation–contraction coupling, as well as Caveolin3 (CAV3) which is implicated in T-tubule biogenesis[22,23]. We showed previously that Mtm1KO

mice display a higher DNM2 level in the symptomatic phase[9]. At 7 weeks of age, Mtm1KO mice TA injected with control ASO presented a higher expression of DNM2 compared to WT confirming an overexpression of DNM2 in the absence of MTM1 and suggesting that the increase in DNM2 is a cause of the disease (Fig. 2a). Treatment with the three ASO candidates reduced DNM2 levels in both WT and Mtm1KO TA. While Mtm1KO TA injected with control ASO were atrophic and displayed a very weak *in situ* muscular force, Mtm1KO TA with *Dnm2* knockdown were remarkably bigger (Fig. 2b,c) and displayed a significant increase in muscle force, which achieved normal levels for ASO #1 (Fig. 2d). Mtm1KO TA injected with control ASO exhibited the typical CNM histology with a predominance of small rounded fibres with mislocalized nuclei and abnormal mitochondrial distribution (Fig. 2e). These abnormal histological features were prevented in Mtm1KO TA injected with the different *Dnm2* ASOs, as reflected by a decrease in the number of fibres with mislocalized nuclei and a restoration of mitochondrial distribution and fibre size (Fig. 2e–g, Supplementary Fig. 1a–c). Altogether, these results suggest that intramuscular injections of ASOs reduce DNM2 levels and prevent myotubular myopathy in Mtm1KO mice.

**ASO systemic injections knockdown DNM2 and prevent CNM.** To assess if systemic ASO delivery can prevent disease progression in this non-dystrophic myopathy and to correlate DNM2 levels with therapeutic effects, a systemic dose response study was performed. ASO#1 was selected due to the strong rescue effects observed when administrated locally into muscle: namely it had the best dose response (Fig. 1b,c) and increased muscle force to normal levels (Fig. 2d). Four escalating doses (3.125, 6.25, 12.5 or $25\,mg\,kg^{-1}$) of ASO#1 were administrated to WT or Mtm1KO mice by intraperitoneal (i.p.) injections once a week starting with mice at 3 weeks of age, and compared to littermates treated with $25\,mg\,kg^{-1}$ control ASO. Mtm1KO mice injected with control ASO presented a short lifespan due to the disease progression and died by about 8 weeks of age, while administration of ASO#1 prolonged the lifespan significantly and in a dose-dependent manner (Fig. 3a). All Mtm1KO mice injected with 12.5 or $25\,mg\,kg^{-1}$ survived until the end of the study at 12 weeks. Notably, administration of 6.25 and even $3.125\,mg\,kg^{-1}$ extended the lifespan of Mtm1KO with a survival rate of 83% and 37%, respectively. In addition, body weight of Mtm1KO mice with control ASO plateaued at 15 g while weekly systemic injections of 3.125 or $6.25\,mg\,kg^{-1}$ of *Dnm2* ASO improved body weight gain up to 18 g, and $25\,mg\,kg^{-1}$ ASO treatment was associated with a normal body weight at 12 weeks (Fig. 3b,c).

Two time points were further evaluated following systemic ASO treatment, at week 7 when some Mtm1KO mice injected with control ASO were still alive, albeit severely affected, and at week 12 to further assess the phenotypes of Mtm1KO with a prolonged lifespan. At week 7, Mtm1KO mice treated with control ASO were severely affected (Supplementary Movie 1) and presented hypotrophic TA and gastrocnemius (gast.) muscles (Fig. 4a). Injections with 12.5 or $25\,mg\,kg^{-1}$ doses of *Dnm2* ASO greatly improved TA and gast. muscle weight. *In situ* TA muscle force was not rescued with 3.125 and $6.25\,mg\,kg^{-1}$ doses but was improved with higher doses, reaching normal levels with $25\,mg\,kg^{-1}$ ASO (Fig. 4b). In addition, while $3.125\,mg\,kg^{-1}$ ASO treatment did not improve fibre hypotrophy, nuclei positioning, or muscle histology relative to Mtm1KO injected with control ASO, $25\,mg\,kg^{-1}$ ASO treatment resulted in a decrease number in fibres with mislocalized nuclei and a strong increase in fibre size and triad ratio (Fig. 4c–e, Supplementary Fig. 2a,b). Mtm1KO injected with control ASO or lower ASO#1 doses exhibited abnormal distribution of

mitochondria oxidative activity; however $25\,mg\,kg^{-1}$ ASO treated Mtm1KO mice presented an amelioration but not a normalization in SDH staining distribution, suggesting that the rescue was not complete at this early stage for this phenotype.

By 12 weeks, all Mtm1KO injected with control ASO died and a dose response correlation in the phenotype rescue was observed with ASO#1 (Fig. 3a–c). Mtm1KO mice injected with $3.125\,mg\,kg^{-1}$ were significantly affected at 11 weeks of age and could not perform the clinical tests such as hanging, grip, string and rotarod tests (Supplementary Fig. 3). Altogether, this low dose did not produce significant clinical and histological improvements; however the partially extended lifespan supports a partial rescuing effect. A significant improvement in performance was observed with 6.25 and $12.5\,mg\,kg^{-1}$ doses. Noteworthy, the highest dose tested greatly improved the ability of Mtm1KO mice to perform comparable to WT in hanging, grip, string and rotarod tests (Supplementary Movie 2, Supplementary Fig. 3), indicating that a dose of $25\,mg\,kg^{-1}$ rescues the whole-body strength. At 12 weeks, this highest dose correlated with a normal specific muscle force and close to normal TA and gast. weight (Fig. 5a,b), muscle histology (with only 4% of fibres with mislocalized nuclei), mitochondria distribution and muscle ultrastructure including rescued triad shape and ratio (Fig. 5c–e, Supplementary Fig. 2c,d). These findings were confirmed by immunolabeling of transversal or longitudinal sections of TA with antibodies against CAV3, DHPR and RYR1. Indeed, myofibers from Mtm1KO mice treated with ASO control presented aggregated or perturbed localization of CAV3, RYR1 and DHPR, while Mtm1KO treated with $25\,mg\,kg^{-1}$ ASO#1 exhibited a normal localization of RYR1 and DHPR with partial decrease of CAV3 aggregates at both 7 and 12 weeks (Supplementary Fig. 4a). ASO#1 injection and DNM2 decrease did not significantly impact on the level of RYR1 nor DHPR (Supplementary Fig. 4b–d). Overall, these results demonstrate that weekly $25\,mg\,kg^{-1}$ ASO systemic delivery to Mtm1KO mice at early disease phase efficiently prevents myopathy progression by rescuing both lifespan and body weight, and correcting muscular mass, force and histology.

In addition to the rescue of locomotor muscles, $25\,mg\,kg^{-1}$ ASO#1 treatment ameliorated greatly the diaphragm muscle histology. At 7 weeks, Mtm1KO mice treated with ASO control presented a very thin diaphragm with mislocalized RYR1 and DHPR proteins as well as disorganized sarcomere ultrastructure (Fig. 6). These features were restored in Mtm1KO treated with ASO#1 at $25\,mg\,kg^{-1}$ at both 7 and 12 weeks. Transversal diaphragm sections presented an increase in muscle thickness due to increase of fibre area and layers composing this muscle, equivalent to WT control mice (Fig. 6a,b). This morphological improvement was accompanied with a normal localization of RYR1 and DHPR within the myofiber as well as sarcomere organization and triad formation that were comparable to WT diaphragms (Fig. 6a). These findings support the clinical improvements of respiratory function and lifespan extension that were observed in Mtm1KO mice treated with $25\,mg\,kg^{-1}$ of ASO#1.

A dose response correlation was noted for the different muscle phenotypes analysed at both ages. This phenotypic amelioration correlated with DNM2 protein levels, while the lower $3.125\,mg\,kg^{-1}$ dose did not lead to a significant DNM2 decrease (Fig. 7a,b). Furthermore, a decrease in DNM2 levels by $\sim50\%$ (obtained with $25\,mg\,kg^{-1}$) is sufficient to achieve disease prevention in Mtm1KO mice, with no impact on lifespan and body weight, muscle force, mass and histology of WT mice (Figs 3–6, and 7a,b, Supplementary Figs 2–4). Interestingly, DNM2 reduction can be achieved after 1 week and maintained at least 2 weeks after a single injection of $25\,mg\,kg^{-1}$ of ASO#1 in WT mice

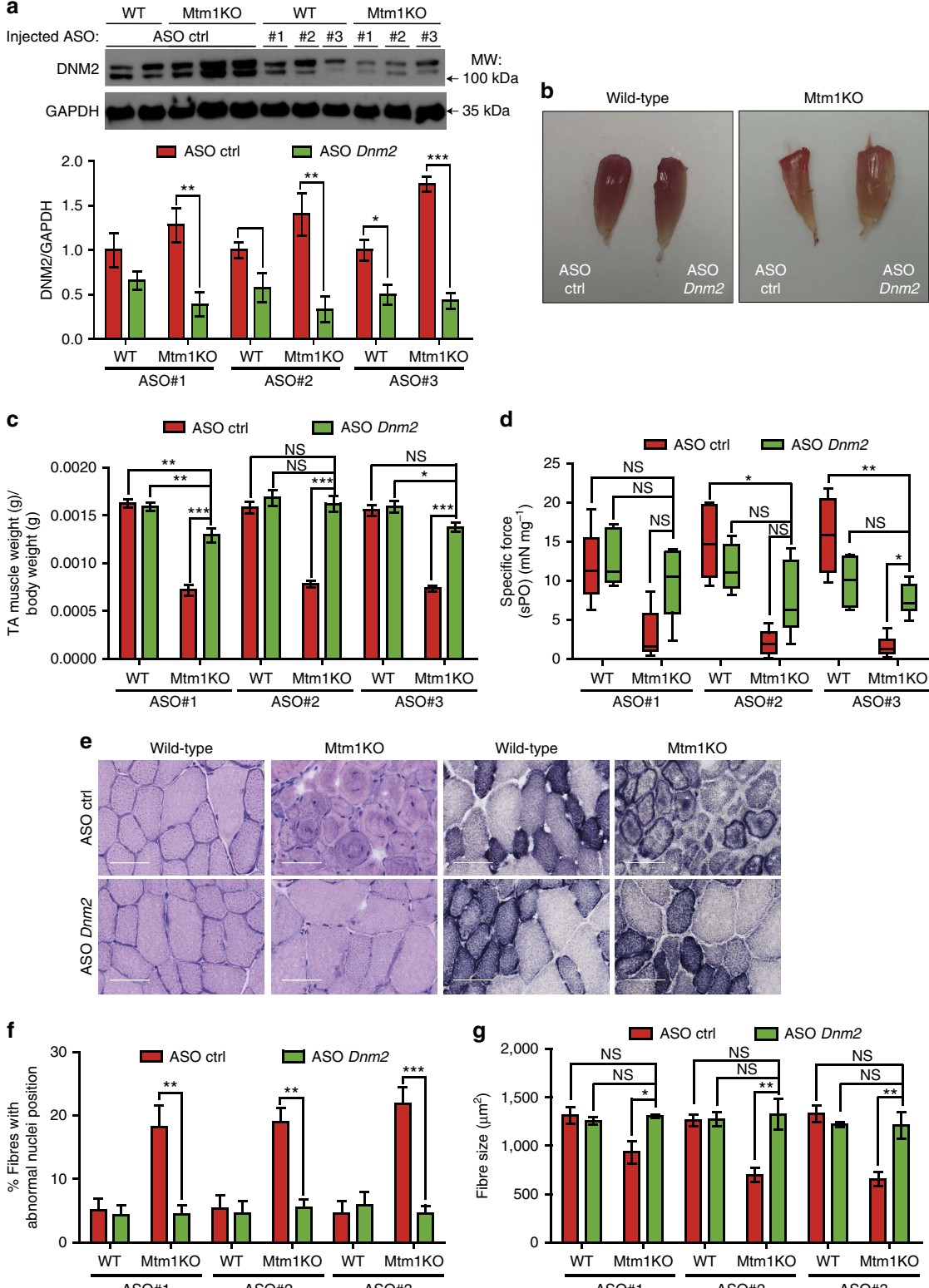

**Figure 2 | *In vivo* validation of ASOs targeting *Dnm2* following intramuscular injection.** (**a**) Representative western blot of 7-week-old WT or Mtm1KO TA muscles injected with 20 μg of ASO control (ctrl), #1, #2 or #3. DNM2 is present as two bands in muscle tissue. DNM2 densitometries were quantified below and standardized to the loading control GAPDH (*n* = 5-7 mice per group). (**b**) Photography of TA muscles from WT or Mtm1KO mice treated with ASO ctrl or ASO *Dnm2* #1. (**c**) TA muscle weight relative to body weight (*n* = 8). (**d**) Specific muscle force of the TA (*n* = 5-6 mice per group).(**e**) TA muscle sections were stained with H&E (left) to visualize nuclei positioning or with SDH (right) for mitochondria oxidative activity distribution. Scale bars, 50 μm. (**f**) Percentage of fibres with mislocalized nuclei (*n* = 4-6 mice per group). (**g**) Fibre area was determined in 600–1,000 fibres per sample (*n* = 5 mice per group). Data represent means ± s.e.m. NS, not statistically significant. *$P < 0.05$, **$P < 0.01$, ***$P < 0.001$ for TA treated with ASO *Dnm2* versus TA treated with ASO ctrl (ANOVA test). kDa, kilodalton; MW, molecular weight.

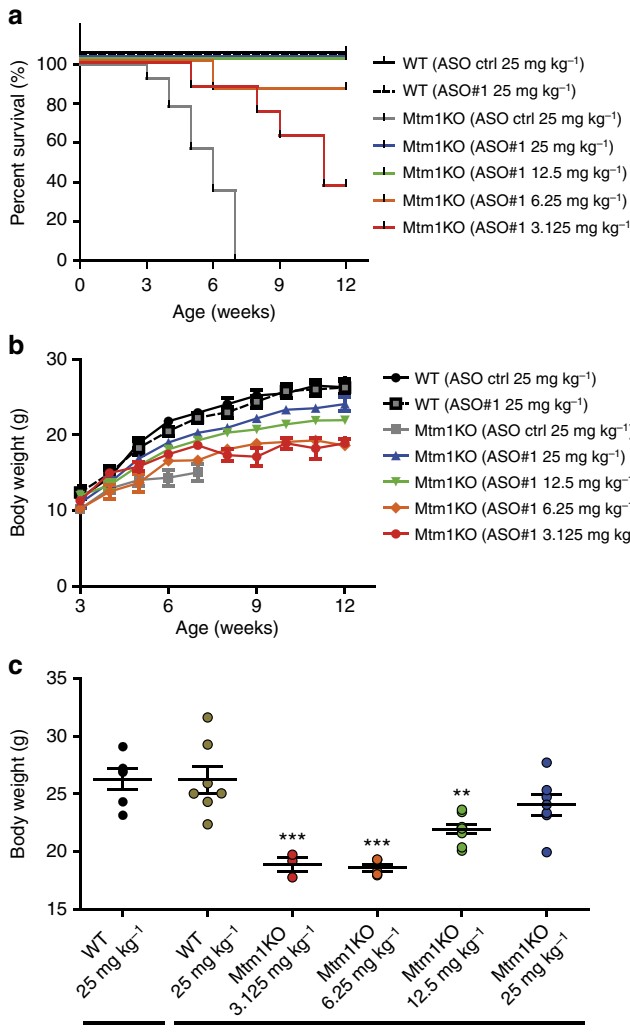

**Figure 3 | Dose–response effect of ASO-*Dnm2* systemic injections.**
Representation of survival percentage (**a**) and whole-body weight evolution (**b**) of WT injected with 25 mg kg$^{-1}$ ASO control (ctrl) or ASO#1, or Mtm1KO injected with 25 mg kg$^{-1}$ ASO ctrl or different doses of ASO#1. ($n = 7$ mice per group at week 3). (**c**) Dots represent individual body weight of ASO-treated WT or Mtm1KO mice at week 12 (sacrifice day). $n = 5$–7 mice per group except for Mtm1KO treated with 3.125 mg kg$^{-1}$ where $n = 3$. **$P < 0.01$, ***$P < 0.001$ for mice treated with ASO#1 versus WT treated with ASO control (ANOVA test followed by *post hoc* Bonferroni).

(Supplementary Fig. 5), suggesting that the applied injection protocol could be ameliorated by decreasing the frequency of the ASO administration and consequently decrease any potential toxicity relative to chronic exposure to ASO. Long-term ASO treatment of certain bicyclic nucleic acids, such as LNAs, have been reported to have an increased risk of liver or kidney toxicity[24–27]. Analysis of histology of Mtm1KO mice weekly injected with 25 mg kg$^{-1}$ ASO for 10 weeks revealed no effects in kidney and liver. Few hepatocytes with condensed nuclei of unknown significance were seen in liver of WT mice only, treated with ASO#1 25 mg kg$^{-1}$. No effect was seen in both Mtm1KO and WT mice on serum biochemical markers of liver and kidney (including aspartate aminotransferase (ASAT), alanine aminotransferase (ALAT), creatinine, urea) (Supplementary Fig. 6). Characterization of ASO distribution in muscles of systemically treated mice demonstrated dose-dependent accumulation of ASO in muscles, supporting the conclusion that ASO efficiently distributes to non-dystrophic muscles (Fig. 7c,d). Unexpectedly, we observed significantly higher distribution of both control ASO and ASO#1 in muscles from the Mtm1KO mice compared to WT, suggesting a selective advantage for ASO-based therapy in this non-dystrophic myopathy. Potentially this observation suggests that lack of MTM1 may increase ASO uptake or decrease ASO degradation in muscle, and that this phenomenon is not reversed upon DNM2 downregulation.

Previous studies revealed abnormalities of neuromuscular junction (NMJ) in a zebrafish model[28] of XLCNM and Mtm1KO mice[29] with enlarged NMJ area. Using a fluorescently labelled bungarotoxin, Acetylcholine nicotinic receptors were labelled to visualize the NMJ of ASO-treated mice. Analysis of the NMJ area revealed that Mtm1KO treated with 25 mg kg$^{-1}$ of either ASO control or ASO#1 similarly exhibited larger NMJ's compared to age-matched WT at both 7 and 12 weeks (Supplementary Fig. 7a,b). Sciatic nerve transversal sections myelination of Mtm1KO appeared normal (Supplementary Fig. 7c). These findings indicate that systemic reduction of DNM2 rescues the CNM phenotype with normal muscle structure and function despite NMJ phenotype persistence, suggesting this NMJ histological phenotype does not have a strong impact on the disease.

**_Dnm2_ ASO reverts CNM phenotypes in affected Mtm1KO mice.**
We next investigated whether ASO systemic delivery can revert CNM features after disease onset. To determine the features that could be reverted, we established a disease severity scoring system (DSS) encompassing six features: body weight difference between Mtm1KO and WT littermates, ability to perform the hanging test, positioning of hindlimbs when walking, ptosis, kyphosis severity, and breathing alteration based on clinical observations (Fig. 8a, Supplementary Table 2). A higher DSS score indicates a more severe phenotype. A large cohort of Mtm1KO mice was followed over time with phenotype progression reaching a mean DSS of 3 at week 5. We therefore selected this age to see if *Dnm2* ASO could revert the disease phenotype. We began to inject 25 mg kg$^{-1}$ ASO#1 once a week. One week later, half of the injected Mtm1KO mice had died, while the other half survived until the end of the study at 12 weeks (Fig. 8b). Retrospectively, the average DSS at week 5 was above 4 for non-rescued mice and 2.5 for the group of rescued mice, suggesting mice that could not be rescued after a single injection were in general more severely affected with more pronounced kyphosis, ptosis and paralysis; however no specific phenotype was predictive of the survival (Fig. 8c,d). After only one injection, surviving Mtm1KO mice presented a stabilization of their phenotype, and were rescued two weeks after commencement of treatment (Fig. 8e,f). The DSS dropped from 2.5 at week 5 to 1 at week 7 (Fig. 8c), indicating a rescue of all clinical features of disease, except for a difference in body weight which persisted until 12 weeks of age (maximum age investigated in this study) (Fig. 8c–e, Supplementary Movie 3). These surviving mice were able to perform hanging, rotarod, string and grip tests at WT levels (Fig. 8f–i). By 12 weeks, they presented a lower muscle mass and force compared to WT (Fig. 9a,b), however, TA muscle fibres had a low number of mislocalized nuclei ($\simeq 6\%$, compared to 27% in 7-week old untreated mice), almost normal fibre size, improvement in mitochondrial distribution and localization and structure of triads (Fig. 9c–g). Delivery of 25 mg kg$^{-1}$ of ASO#1 from 5 weeks of age reduced DNM2 to normal levels (Fig. 9h). Taken together, these results show that post-symptomatic ASO treatment quickly reverts almost all CNM phenotypes within 2 weeks, with disease amelioration after a single injection.

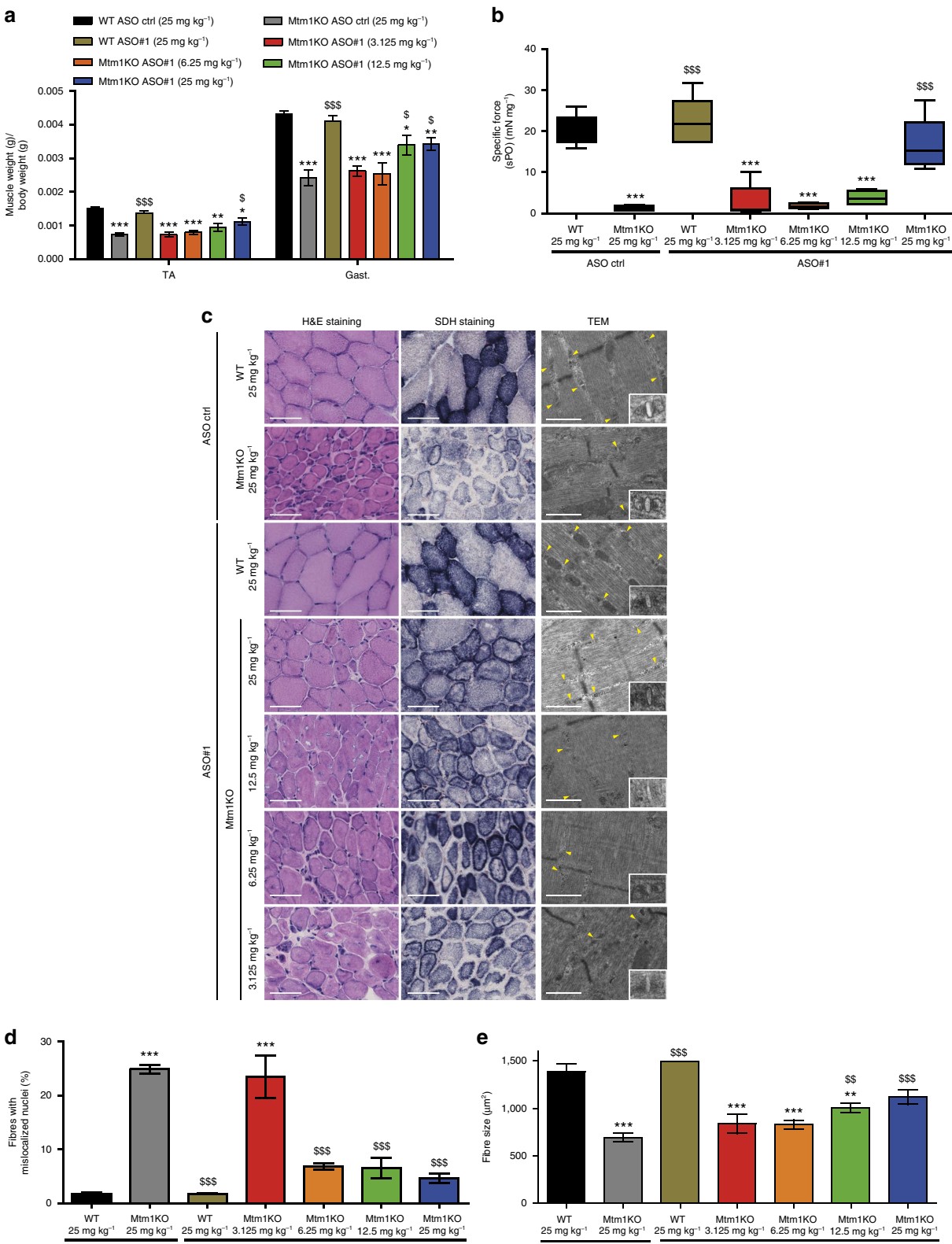

**Figure 4 | Physiological effects of ASO-*Dnm2* systemic injections at 7 weeks of age.** (**a**) Following 5-weekly injections of ASO, mice were killed and TA and gast. muscles were weighed (*n* = 5–7 mice per group). (**b**) TA specific muscle force was measured after sciatic nerve stimulation. The specific muscle force was calculated by dividing the absolute force by the TA weight (*n* = 5–7 mice per group). (**c**) TA muscle sections were stained for H&E or SDH. Sarcomere and triads (yellow arrows) ultrastructure was assessed by TEM. Scale bars: 50 μm (H&E and SDH) or 500 nm (TEM) images. (**d**) Percentage of fibres with mislocalized nuclei was determined in 1,000 fibres (*n* = 5–6 mice per group). (**e**) TA muscle fibre area was calculated on 300–600 fibres per sample (*n* = 5 mice per group). Data represent mean ± s.e.m. \*$P < 0.05$, \*\*$P < 0.01$, \*\*\*$P < 0.001$ for mice treated with ASO *Dnm2* versus WT treated with ASO ctrl. $^{\$}P < 0.05$, $^{\$\$}P < 0.01$, $^{\$\$\$}P < 0.001$ for mice treated with ASO *Dnm2* versus Mtm1KO treated with ASO ctrl (two-way ANOVA followed by *post hoc* Bonferroni).

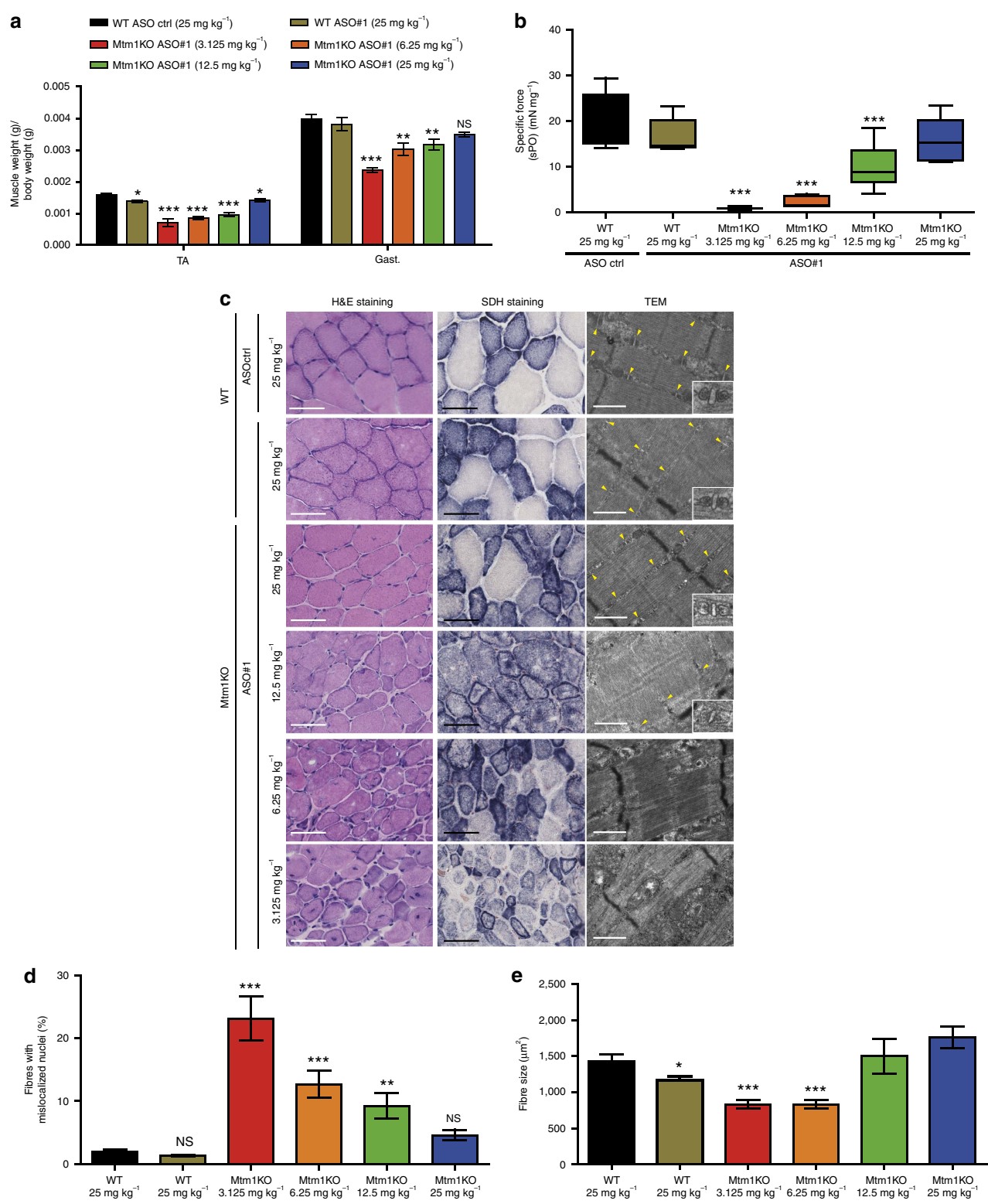

**Figure 5 | Physiological effects of ASO-*Dnm2* systemic injections at 12 weeks of age.** (**a**) TA and gast. muscle weights from these mice were measured and the ratio relative to the body weight was represented. (**b**) The TA *in situ* muscle force was measured after sciatic nerve stimulation. The specific muscle force was calculated by dividing the absolute force by the TA weight. (**c**) TA muscle transversal sections stained with H&E (left panel) and SDH (middle panel) at 12 weeks (10 weeks of treatment). Sarcomere and triads (yellow arrows) ultrastructure assessed by TEM (right panel). Scale bars for H&E and SDH is 50 μm and 500 nm for TEM pictures. (**d**) The percentage of fibres with mislocalized nuclei was counted on 1,000 fibres per group. (**e**) TA muscle fibre area was calculated on 300–600 fibres per sample. For all tests cited above, $n = 5$-7 mice per group except for Mtm1KO treated with 3.125 mg kg$^{-1}$ where $n = 3$. Data represent means ± s.e.m. NS, no statistical significance, $*P < 0.05$, $**P < 0.01$, $***P < 0.001$ for mice treated with ASO#1 versus WT treated with ASO control (ctrl) (ANOVA test followed by *post hoc* Bonferroni).

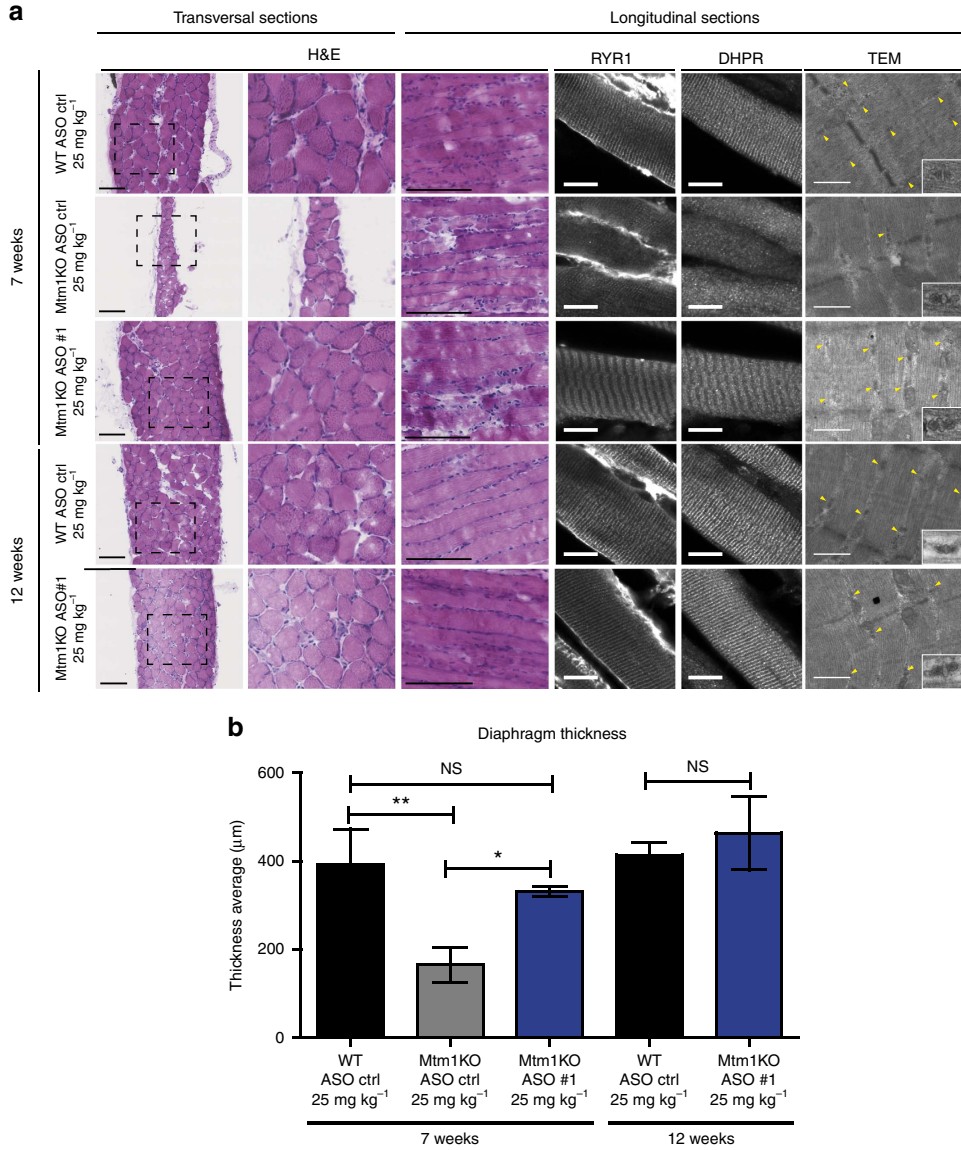

**Figure 6 | Diaphragm histology and ultrastructure organization.** (**a**) Hematoxylin and eosin (H&E) staining of cross or longitudinal diaphragm sections of WT or Mtm1KO mice treated with ASO control (ctrl) or ASO#1 at different ages (7 or 12 weeks). Scale bars, 100 μm. These longitudinal sections have been immunostained with antibodies against RYR1 and DHPR. Scale bars, 10 μm. The ultrastructure of diaphragm was analysed using electronic microscopy to assess the sarcomere organization and triad formation and shape (yellow arrows). Scale bars, 500 nm. (**b**) Graph depicting mean ± s.e.m. of quantification of diaphragm transversal sections thickness. $n = 4$-5 samples. NS, not statistically significant, $^*P < 0.05$, $^{**}P < 0.01$ (ANOVA test followed by *post hoc* Bonferroni).

## Discussion

In this study we demonstrated that DNM2 knockdown, through ASO delivery, prevents myotubular myopathy in Mtm1KO mouse model by extending the lifespan and restoring muscle force, mass and histology in a dose-dependent manner. Weekly administration of 25 mg kg$^{-1}$ at early disease stage rescues most myopathy features by 12 weeks. However, long-term effects of this approach are unknown. In addition, DNM2 is ubiquitously expressed and plays an important role in many cellular processes such endocytosis, intracellular membrane trafficking and cytoskeleton organization[30]. It is important to note that *Dnm2* heterozygous mice do not present any particular phenotype[9]. However, determination of side effects related to DNM2 reduction through ASO is needed to confirm the safety of this approach.

Furthermore, a single ASO administration into affected Mtm1KO mice stabilizes the CNM phenotype progression and quickly reverts it following the second injection. However, some

mice that exhibited a very severe phenotype with a higher DSS at the start of the injections were not able to be rescued. This observation suggests that disease reversion can be better obtained when ASO treatment is initiated at earlier disease stages. On the other hand, ASO activity to decrease DNM2 and reverse the phenotype appears to have a short delay as the disease can be stabilized one week after the first injection and significantly reversed after 2 weeks of treatment. Moreover, while DNM2 protein level is increased in the disease state, disease prevention with 12.5 mg kg$^{-1}$ (Fig. 7b) and disease reversion with 25 mg kg$^{-1}$ (Fig. 9h) both correlated with DNM2 level normalization, suggesting it is not necessary to decrease DNM2 below normal level to reach a benefit while minimizing the potential risk of DNM2 target reduction.

This study confirms the epistasis between *Mtm1* and *Dnm2* and the 'cross-therapy' rationale that targeting another myopathy gene (*DNM2*) than the one mutated in the disease (*MTM1*) can

efficiently re-balance muscle function. To date no direct link has been established between MTM1 and DNM2. MTM1 is involved in the regulation of the phosphoinositides phosphorylation and thus potentially in regulation of the physicochemical properties of intracellular membranes, while DNM2 plays an important role in membrane remodelling. We hypothesize that in absence of MTM1, DNM2 level and activity should be normalized to maintain the cellular homeostasis.

This first pharmacological validation of DNM2 as a therapeutic target through ASO-mediated knockdown provides an attractive therapeutic strategy that may be applied to patients with this severe congenital myopathy. Nonetheless, these data were generated using ASOs against the murine DNM2 in the mouse model of XLCNM. Moving towards clinical trials will require the development of a potent human ASO compound with an acceptable safety profile. Identification and characterization of the human candidate including off-target analysis should be completed in human cells.

In conclusion, while various ASO chemistries may differ in their ability to achieve significant target reduction and disease rescue in skeletal muscle, our data reveal that the advanced cEt ASO chemistry used here appears promising for non-dystrophic muscle diseases.

## Methods

**Animals.** Mtm1KO or WT 129SvPAS mice were generated by crossing *Mtm1* heterozygous females obtained by homologous recombination[17] with WT males. Mice were handled according to the French and European legislation on animal care and experimentation. Protocols were approved by the institutional Ethics Committee. Protocols No.:Com'Eth IGBMC-ICS 2012-132, 2013-034 and 2016-5453 were granted to perform animal experiment. Mice were kept on 12 h day light and 12 h cycle and given free access to standard food. Lifespan and body weight were followed during this study. All mice analysed in this study were male.

**Antisense oligonucleotides.** All ASO used in these studies were synthesized in IONIS Pharmaceuticals. They were 16 nucleotides in length and chemically modified with phosphorothioate in the backbone and cEt modifications on the wings with a deoxy gap (3-10-3 design). Oligonucleotides were synthesized using an Applied Biosystems 380B automated DNA synthesizer (PerkinElmer Life and Analytical Sciences-Applied Biosystems, Waltham, Massachusetts) and purified[15]. A total of 500 ASOs were prescreened in b.END cells. The three best ASO candidates that reduce *Dnm2* level have been selected. In addition, a random control ASO sequence was used as control. The sequence of each ASO is listed in Supplementary Table 1. The chemical structures in Supplementary Table 1 were drawn using ChemDraw software.

**ASO quantification in muscle, liver and kidney.** Samples and calibration standards were aliquoted into 96-well plates and internal standards were added. Aliquots in 96-well plates were extracted via a liquid–liquid extraction using ammonium hydroxide and phenol:chloroform:isoamyl alcohol (25:24:1). The aqueous layer was further processed via solid phase extraction (Phenomenex Inc., Strata X SPE), dried under nitrogen and reconstituted in 120 µl of water containing 100 µM EDTA. Samples were injected into an Agilent 6460 Triple Quad LC/MS system for analysis. The calibration range for control ASO was 0.005–30 µM (0.027–163 µg g$^{-1}$) in 50 mg mouse liver homogenate; the calibration range for ASO#1 was 0.005–30 µM (0.028–165 µg g$^{-1}$) in 50 mg mouse liver homogenate.

**Cell transfection.** C2C12 mouse myoblasts were purchased from ECACC. They were electroporated with different concentrations of ASO using Amaxanucleofector2B Kit V and following the manufacturer's instructions. Briefly, cells were trypsinized and $1 \times 10^6$ cells per sample were gently centrifuged at 90 *g* for 10 min at room temperature. Cell pellet was resuspended in 100 µl of the solution provided in the kit. Then, 0.015, 0.06, 0.25 or 1 µM of ASO control, #1, #2 or #3

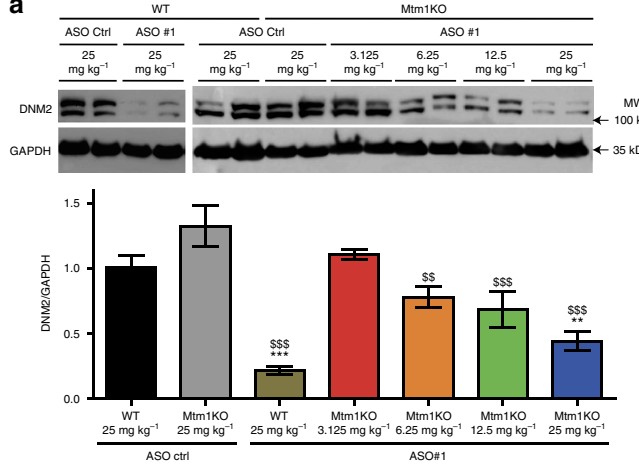
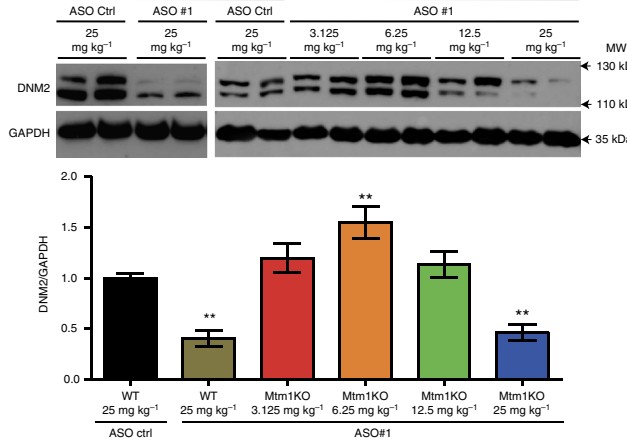
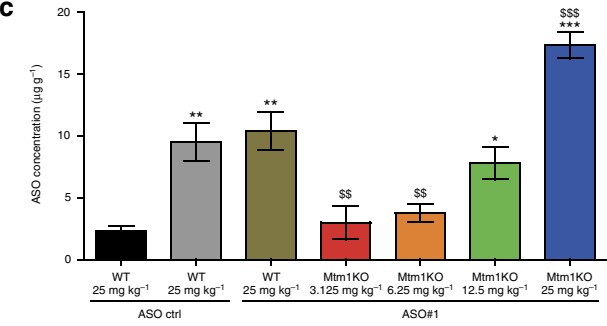
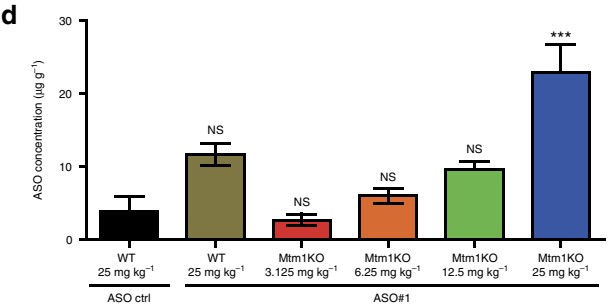

**Figure 7 | DNM2 knockdown and ASO concentration in skeletal muscles.** (**a,b**) DNM2 and GAPDH (loading control) expression in TA muscles of ASO-treated WT or Mtm1KO mice at 7 weeks (**a**) (*n* = 6–7 mice per group) or at 12 weeks of age (**b**) (*n* = 5–7 mice per group except for Mtm1KO treated with 3.125 mg per kg where *n* = 3). (**c,d**) ASO concentration was determined in gast. muscle of 7- (**c**) or 12- (**d**) week-old mice using mass spectrometry and normalized to muscle weight (*n* = 4 mice per group). Data represent mean ± s.e.m. NS, not statistically significant, *$P < 0.05$, **$P < 0.01$, ***$P < 0.001$ for mice treated with ASO *Dnm*2 versus WT treated with ASO ctrl. $^{\$\$}P < 0.01$, $^{\$\$\$}P < 0.001$ for mice treated with ASO *Dnm*2 versus Mtm1KO mice treated with ASO ctrl (two-way ANOVA followed by *post hoc* Bonferroni). kDa, kilodalton; MW, molecular weight.

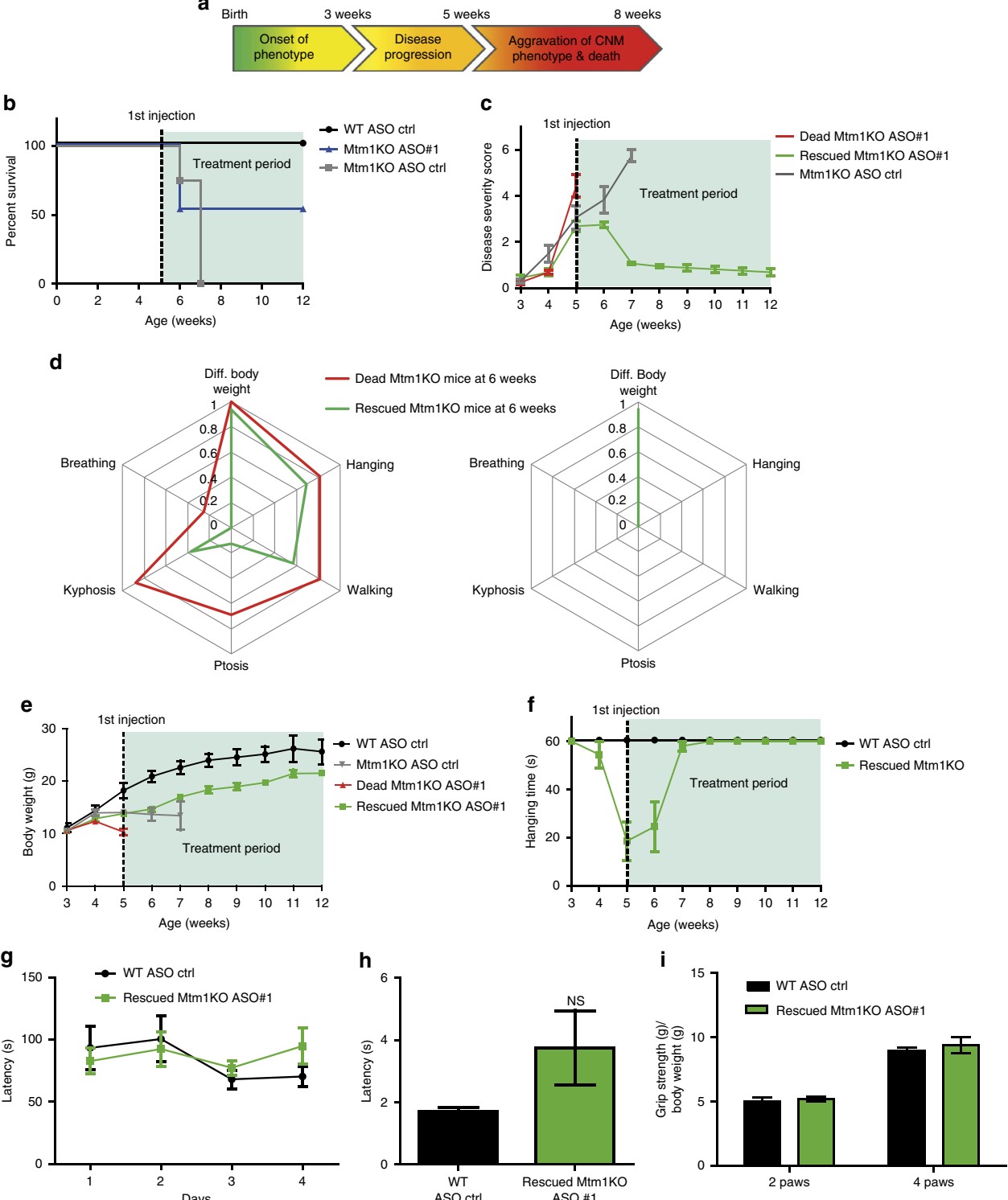

**Figure 8 | ASO-*Dnm2* systemic injections revert installed muscle defects in Mtm1KO mice.** (**a**) Chronology of CNM phenotype onset and evolution in Mtm1KO mice. (**b**) Survival of WT (*n* = 5) or Mtm1KO (*n* = 16) mice upon ASO treatment started at 5 weeks. (**c**) Disease severity score (DSS) evolution of Mtm1KO mice that died during the first week of treatment or have been rescued after ASO treatment started at week 5. (**d**) Radar chart representing average values of the six main CNM features in Mtm1KO mice at 5 weeks old (before ASO treatment, left panel) and their evolution at week 12 (after 8 injections, right panel). (**e**) Body weight evolution of WT treated with ASO control (*n* = 5), Mtm1KO mice treated with ASO control (*n* = 8) or ASO#1 (*n* = 8 for dead or rescued Mtm1KO groups). (**f**) Hanging test performance of WT (*n* = 5) or Mtm1KO (*n* = 8) rescued by ASO treatment. (**g-i**) Clinical tests done at 11 weeks of age, WT or Mtm1KO mice underwent different clinical tests to assess whole body strength, fine motor coordination, balance and resistance to fatigue using rotarod (**g**), string test (**h**) and grip test (**i**). NS, not statistically significant (*t*-test).

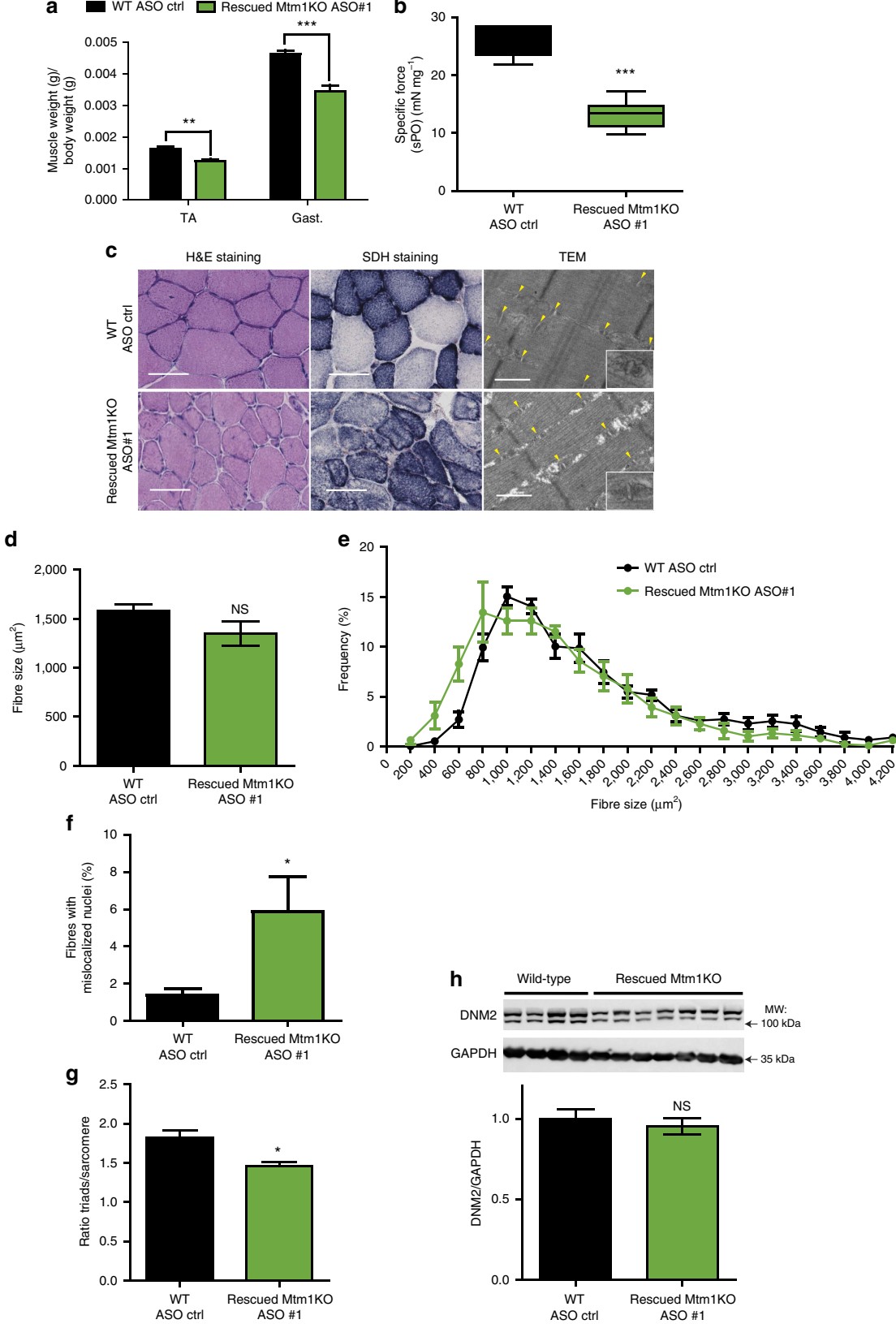

**Figure 9 | Physiological improvements of CNM phenotype in severely affected Mtm1KO mice.** (**a**) TA and gast. muscles weight and (**b**) *in situ* specific muscle force at week 12 and following 8 weeks of ASO treatment. (**c**) TA muscle sections were stained with H&E or SDH. Scale bar, 50 μm for histology and 500 nm for TEM. Yellow arrows point to organized triads. (**d**) Fibre size average was quantified in 400–600 fibres ($n = 5$). (**e**) TA muscle fibre area distribution of Mtm1KO mice injected with ASO#1 versus WT injected with ASO control ($n = 5$). (**f**) Fibres with abnormal position of nuclei ($n = 5$). (**g**) Ratio of triads per sarcomere. (**h**) DNM2 and GAPDH (loading control) protein levels were measured in TA muscles by western blot. Data represent means ± s.e.m. NS, not statistically significant, *$P < 0.05$, **$P < 0.01$, ***$P < 0.001$ (*t*-test). kDa, kilodalton; MW, molecular weight.

was added to the cell suspension and transferred immediately to an electroporation cuvette. Electroporation was performed using the B-032 programme of Nucleofector I Device. The electroporated cells were incubated overnight in DMEM medium containing 20% FCS at 37 °C, before collecting of the cells for RNA and protein analysis.

**ASO intramuscular injections.** Right TA muscles of Mtm1KO or WT mice were injected with 20 μg ASO Dnm2, while contralateral TA were injected with 20 μg of ASO control. Mice were injected from week 3 to week 7 of age under isoflurane anaesthesia and were killed 2 days after the last injection.

**Intraperitoneal injection.** All ASO were dissolved in filtered and autoclaved sterile PBS. Doses of 3.125, 6.25, 12.5 or 25 mg kg$^{-1}$ were injected by i.p. once a week into WT or Mtm1KO. Mice were killed 2 days after the last injection for analysis.

**Clinical tests.** *Grip test.* Mice were placed on the wire grid of the grip-strength apparatus which was connected to an isometric force transducer (dynamometer). They were lifted by the tail so that their all paws grasp the grid and they were gently pulled backward by the tail until they release the grid. The maximal force exerted by the mouse before losing grip was recorded. The mean of three measurements for each animal was calculated. Results are represented relative to whole body weight.

*String test.* Mice were placed on a wire held under high tension between two vertical supports and elevated 40 cm from the flat surface. The latency of a mouse to lift and grab the string with hind limbs was measured for each mouse over three trials with inter-trial intervals of 5 min between each trial.

*Whole body hanging test.* Mice were placed on a grid (cage lid) then turned upside down; the suspending animal should hold on to the grid to avoid falling. The latency to fall was measured three times for each mouse. The three trials were taken at ten minute intervals to allow a recovery period. The latency time measurements began from the point when the mouse was hanging free on the wire and ended with the animal falling to the cage underneath the wire or grid. The maximum time measured was 60 s. For all physiological tests described above, two trials were done to familiarize the mouse with the testing conditions. The data were expressed as an average of three trials.

*Rotarod test.* Motor coordination, balance and resistance to fatigue were assessed with a rotarod apparatus. At day 0, mice were trained to walk against the motion of a rotating drum. First, a training session of 5 min was done at a constant speed (4 r.p.m.; rotation per minute) followed by three trials at an accelerating speed. The following days, mice were tested three times at an accelerating speed (from 0 to 40 r.p.m. over 5 min). The mean latency to fall off the rotarod was recorded. The test was repeated three times for four consecutive days (day 1–4).

**TA muscle contractile properties.** TA muscle contraction properties were evaluated *in situ* after sciatic nerve stimulation using Aurora scientific force transducer. Briefly, mice were anaesthetised with i.p. injection of pentobarbital (60 mg kg$^{-1}$). The distal tendon of TA muscle was detached and tied to the isometric transducer. The absolute maximal force was measured after stimulation of the sciatic nerve by pulses of 50–150 Hz. The specific maximal force was determined by dividing the absolute muscle force on the TA muscle weight.

**Tissue collection.** Mice were killed by cervical dislocation after carbon dioxide (CO$_2$) suffocation. The TA and gast. muscles were dissected and weighed. They were snap-frozen in liquid nitrogen-cooled isopentane and stored at $-80$ °C for H&E and SDH histology analysis. For immunostaining, TA and diaphragm muscles were stored in PFA 4% for 24 h, then they were transferred to sucrose 30% overnight and stored at 4 °C. Liver and kidney were collected and stored in 4% PFA for 24 h then in 70% ethanol for histological analysis. Diaphragms were embedded in optimal cutting temperature compound. For transmission electron microscopy (TEM) analysis and toluidine blue staining, TA and non stimulated sciatic nerves were dissected and fixed in 4% PFA and 2.5% glutaraldehyde in 0.1 M.

**Disease severity scoring system.** A scoring system was set up to evaluate the clinical evolution of six CNM features. Difference of body weight versus WT littermate, ability to perform the hanging test, walking manner, presence or absence of ptosis and kyphosis and breathing difficulties (frequency and amplitude evaluation based on clinical observations) have been followed every week (before and after ASO treatment) and a score of 0, 0.5 or 1 was given to each clinical readout. The sum represents the DSS. The higher the DSS, the more severe the phenotype, minimum 0 (healthy mouse), maximum 6 (severely affected mouse).

**Transmission electron microscopy.** TEM was carried out on TA muscles stored in 2.5% paraformaldehyde, 2.5% glutaraldehyde and 50 mM CaCl$_2$ in 0.1 M cacodylate buffer (pH 7.4). Sections (70 nm) were stained with uranyl acetate and lead citrate and examined by TEM (Morgagni 268D, FEI). The ratio of triads to sarcomere was calculated by dividing the number of triads identified by the number of sarcomeres present in the field.

**Histological staining.** Transversal or longitudinal cryosections (8 μm) of TA and diaphragm or 5 μm from paraffin-embedded liver or kidney were prepared, fixed and stained by Haematoxylin and Eosin (H&E) or succinate dehydrogenase (SDH). Sections were imaged with the Hamamatsu NanoZoomer 2HT slide-scanner. The percentage of TA muscle fibres with centralized or internalized nuclei was counted using the cell counter plugin in Fiji image analysis software. The fibre area was measured using the Fiji software.

**TA and diaphragm immunostaining.** Transversal or longitudinal cryosections (8 μm) of TA and diaphragm were stained with antibodies against CAV3 (Santa Cruz N-18 sc-7665; 1:500), RYR1 (34C abcam2668; 1:500), DHPR 1 A (abcam2862; 1:500) or BiotiumCF488A α-bungarotoxin (1:1,000). Images were taken in the same Leica SP8-UV confocal microscope. NMJ area was measured using Fiji software.

**RNA extraction and qRT-PCR.** Total RNA was isolated from electroporated cells or muscle tissue using TRIzol reagent according to the manufacturer's instruction (Invitrogen, UK). RT-PCR was carried out on 1–1.5 μg aliquot using SuperscriptII Reverse Transcriptase (Thermofischer Scientific). qRTPCR was performed in Lightcycler 480 (Roche) using: Dnm2 (F): CCAACAAAGGCATCTCCCCT, Dnm2(R):TGGTGAGTAGACCCGAAGGT, Hprt(F): GTAATGATCAGT-CAACGGGGGAC and Hprt (R): CCAGCAAGCTTGCAACCTTAACCA mixed in SybrGreen (Qiagen).

**Protein extraction and western blot.** TA muscle cryosections, C2C12 cells, liver or kidney samples were lysed in RIPA buffer supplemented with PMSF 1 mM and complete mini EDTA-free protease inhibitor cocktail (Roche Diagnostic). Protein concentrations were determined with the BIO-RAD Protein Assay Kit. Samples were denatured at 95 °C for 5 min. Then 20 μg of protein was loaded in buffer containing 50 mM Tris–HCl, 2% SDS, 10% glycerol, separated in 10% SDS–polyacrylamide gel electrophoresis electrophoretic gel and transferred on nitrocellulose membrane for 1.5 h at 200 mA. Membranes were blocked for 2 h in TBS containing 5% non-fat dry milk and 0.1% Tween20 before an incubation for 2 h with primary rabbit polyclonal antibodies against DNM2 2865 (1:500), mouse antibody against RYR1 (34C abcam2668; 1:500), mouse antibody against DHPR 1A (abcam2862; 1:500), homemade mouse antibody against BETA TUBULIN (1:1,000) and mouse antibody against GAPDH (1:100,000) diluted in blocking buffer containing 5% milk. Secondary antibodies coupled to horseradish peroxidase were goat anti-rabbit (for DNM2) (1:10,000) or goat anti-mouse (for RYR1, DHPR, BETA TUBULIN and GAPDH) (1:10,000) and were incubated overnight. Nitrocellulose membranes were visualized in Amersham Imager 600. Full blots of all western blots are presented in Supplementary Fig. 8.

**Blood sample collection and biochemistry analysis.** Blood samples were collected by cardiac puncture from anaesthetised mice. ASAT, ALAT, urea and creatinine levels were determined using an Olympus analyzer with kits and controls supplied by Olympus or other suppliers.

**Statistical analysis.** All data are expressed as mean ± s.e.m. Graphs and curves were made using GraphPad Prism software. Difference between two groups was analysed by *t*-test. For comparison between three groups, two-way ANOVA followed by *post hoc* Bonferroni was used.

**Data availability.** All relevant data that support the findings of this study are available from the corresponding authors upon reasonable request.

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

## Acknowledgements

We thank Pascal Kessler, Olivia Wendling, Huges Jacobs, William Magnant and IONIS PK team for the excellent technical assistance. This work was supported by SATT Conectus Alsace, the Agence Nationale de la Recherche grant ANR-10-LABX-0030-INRT. H.T. was supported by a MRT fellowship.

## Author contributions

B.S.C. and J.L. conceived the project. H.T., B.S.C. and J.L. designed the experiments, analysed the data. H.T. wrote the manuscript with input from all coauthors. B.S.C. and J.L. edited and corrected the manuscript. S.Gu., S.B., S.Gr. and B.P.M. performed the *in vitro* and *in vivo* ASO screen and PK analysis. H.T. carried out most experiments. N.M. performed electron microscopy. S.B. and C.K. performed mice phenotyping.

## Additional information

**Competing interests:** H.T., B.S.C. and J.L. are co-inventors of the patent on therapies targeting dynamin2 to rescue centronuclear myopathies. The other authors declare no competing financial interests.

**Publisher's note**: 

