## [Peer review file · Nature Communications]

Reviewers' comments:

Reviewer #1 (Remarks to the Author):

This is an elegant study with interesting results and potential for translation into a human therapy. I have a few comments only:

1. The authors should explain why the overexpression of DNM2 is deleterious (what is the mechanism) when associated with MTM1 mutations and also why it does not represent a protective/adaptive mechanism to mutations in MTM1.
2. Could the proposed treatment - MTM1 reduction have a deleterious effect on a long term?
3. As MTM1 is expressed in other tissues what happened to the mice injected with the ASO#1; for instance on what regards the blood cells, nerve or even the NMJ?
4. What happened to the WT mice injected with the ASO#1? There is no description of it?
5. It should be explained why the authors chose one ASO over the others for the systemic studies.

Reviewer #2 (Remarks to the Author):

The work presented by Tasfaout et al. extends a previous study by Dr. Laporte's laboratory showing that dynamin-2 is upregulated in myotubular myopathy and that reducing dynamin-2 expression is sufficient to improve XLCNM. The current work builds on this idea and now utilizes anti-sense oligonucleotide technology to reduce dynamin-2 expression and prevent myotubular myopathy in a mouse model. This manuscript evaluates a large number of physical and histological parameters and shows that a reduction in dynamin-2, through ASO delivery, is sufficient to extend the lifespan of Mtm1KO mice and significantly improve muscle strength. This manuscript is of high technical quality, well constructed, and should interest a large number of readers.

- 1) The DSS score includes breathing rhythm in the scoring system. What parameters were used to measure respiration? Does ASO treatment improve these respiratory parameters in Mtm1KO mice? Additionally, in diaphragm muscle, is there a similar improvement in histology, T-tubule morphology, and force with systemic ASO treatment?
- 2) Caveolin-3 aggregates in myotubular myopathy. Does ASO treatment ameliorate CAV3 aggregation in myofibers? RyR and DHPR protein expression are decreased with disease progression and are also mislocalized. Does ASO injection rescue this phenotype?
- 3) Does systemic ASO#1 treatment, injected after disease onset (Figure 4), also improve the triad/sarcomere ratio and T-tubule morphology?
- 4) Could the authors comment on any off target effects of DNM2 reduction as DNM2 is ubiquitously expressed or if they have experienced detrimental effects with decreasing DNM2 beyond a certain threshold?
- 5) What was the level of DNM2 reduction in the severe, dead mice?
- 6) Figure 3a and 3b show data from WT+25mg/kgASO#1. The data from this cohort is not included in 3c-i. Were there any significant effects noted in this cohort of mice for any of the parameters analyzed in 3c-i?
- 7) In Figure 3, 25mg/kg ASO injections resulted in DNM2 levels that were 50% of wildtype levels. In Figure 4, this dosage, starting two weeks later, resulted in DNM2 levels equivalent to wildtype. What is the timing of DNM2 reduction with ASO injection? Please explain this difference in DNM2 expression.

Reviewer #3 (Remarks to the Author):

Myotubularin is a lipid phosphatase centrally involved in muscle membrane homeostasis, as shown also by the fact that mutations in the human MTM1 gene cause myotubular myopathy, a severe recessive X-linked disease of the centronuclear myopathy group with early onset and premature death. CNM can also be caused in dominant form by mutations in dynamin-2 (Dnm2), a GTPase that is intriguingly involved in vesicular traffic or microtubule dynamics.

This study builds on earlier observations that the double knockout of myotubularin and dynamin-1 (Mtm1, Dnm2) rescues the myopathic phenotype caused by Mtm1 deficiency alone. The authors, by translating these findings to an antisense oligonucleotide-based down-regulation of Dnm2, significantly show that this Dnm2 reduction can prevent and even revert myotubular myopathy symptoms in the Mtm1 mouse model in a dose-dependent way, and thus open the door to a potential therapeutic approach for this rare but devastating myopathy.

These results are very impressive- all quantitative parameters show a dramatic improvement of morphology (including correction of the central feature of myotubular myopathy, the centronucleation) and functional performance, as well as a significant extension of life-span in those mice that survive beyond the first week of treatment after age 5 weeks.

Apart from this translational angle, the work also firmly supports the mechanistic interplay between Dnm2 and Mtm1 in basic muscle biology and is therefore also of fundamental interest.

This is overall a well-written and illustrated, experimentally and conceptually excellent paper that offers both intriguing biological and pathomechanistic insight. There are some questions that the authors should address, however.

Comments:

1. Please state the gender of the animals used in the experiments!
2. The GAPDH control blot in Figure 3h seems identical to that in Supplemental Figure 6, yet the dosage regimes and genotypes would be different.
3. Are the differences in survival in Figure 4b statistically significant?
4. Specific force in WT control mice in Figure 3d is somewhat lower than in 4g- please explain the difference (age?)
5. What is the explanation why in Figure 4k there is no decrease in Dnm2 below the normal levels compared to WT?
6. It would be good to indicate the molecular weights for the bands in the Western blots of the figures.

Reviewer #4 (Remarks to the Author):

In this paper, Tasfaout et al. examined the effects of antisense-oligonucleotide-mediated knockdown of Dnm2 in a mouse model of myotubular myopathy. The author found that the novel oligonucleotides significantly reduce the expression of Dnm2, accompanied by the ameliorated pathology in mutant mice. The finding is interesting and promising. It will provide meaningful information that is of use in designing therapeutic strategy. However, there are many flaws and points that should be addressed to strengthen author's claim.

Main Points

1. No toxicology data is included. Blood biochemistry (e.g. BUN, AST, ALT etc.) and histology of kidney and liver should be included. Some oligonucleotides (e.g. 2'MOE) accumulate in kidney, and this possibility needs to be examined.
2. Line 95- The authors' statement that injections with 6.25...25 mg/kg doses improved muscle weight is not supported by the data. The figure shows no statistical significance except for the 25 mg/kg group (fig 3c).
3. Line 142- The authors' claim that the mice could not be rescued by a single injection contradicts with the statement in line 145 that after only one injection mice improved quickly. From the figure, it appears multiple injections are needed.
4. In several figures, the t-test is used inappropriately (e.g. fig 1, 2, suppl. fig 3). ANOVA should be used to compare multiple groups.
5. In addition, in several figures (e.g. fig 1), the comparison is made against non-treated groups, however, they should be compared against ctrl ASO.
6. The authors examined the effects only up to 12 weeks. How long does the effect persist?
7. Line 129- : The authors claim that a significantly higher distribution was observed with both control ASO and ASO #1 in muscles from the mutant mice compared to WT, however, this is not supported by the data. The figure does not show statistics. In addition, authors do not explain the mechanisms underlying the differences.
8. In figure 1 and 2, the number of experiments is not shown.
9. In Figure 3h, there are two bands that are unexplained.
10. The histology data (figure 3e) are too small and hard to evaluate.
11. Figure 4 J and K: The authors claim that no significant differences were observed between rescued mice and WT but no detail about the statistical analysis is given.

Minor Points

1. Apparently, the authors employ novel antisense oligos but the manuscript is hard to follow without the structure of the oligonucleotide.
2. There should be a space between the numerical value and unit symbol

Reviewer #1 (Remarks to the Author):

This is an elegant study with interesting results and potential for translation into a human therapy. I have a few comments only:

1. The authors should explain why the overexpression of DNM2 is deleterious (what is the mechanism) when associated with MTM1 mutations and also why it does not represent a protective/adaptive mechanism to mutations in MTM1.

- It has been shown that overexpression of wild-type DNM2 into muscles of wild-type mice using AAV induces structural defect and muscle weakness similar to centronuclear/myotubular myopathy (Cowling et al. 2011). In addition, mice transgenic for human DNM2 overexpression similarly display a centronuclear/myotubular myopathy phenotype (Liu et al. 2011). In the other hand, it appears that symptomatic *Mtm1* knockout mice present an increased level of DNM2, while reduction of DNM2 in *Mtm1*KO mice rescues myotubular myopathy with correction of muscle force, weight and histology (Cowling et al. 2014; and this study). Taking together, these findings support that DNM2 overexpression is deleterious and a consequence of MTM1 absence and its reduction would present a possibility to restore muscle dysfunction. These points have been stated in the manuscript text (line 37- 42). We also added in the introduction that “autosomal dominant CNM forms are due to heterozygous mutations in *DNM2* that were proposed to increase the GTPase activity and oligomerization of DNM2” (text line 37- 42) referring to two in vitro studies from Lemmon and Albanesi’s labs. It is not clear at present how increased DNM2 function would promote nuclei misposition, triad defects and muscle weakness, and will be the aim of future studies. The fact that increased DNM2 level does not represent a protective mechanism may reveal either that MTM1 is directly controlling DNM2 expression or stability, or an unforeseen complex regulatory mechanism (eg. DNM2 needs to be in a special state to protect).

2. Could the proposed treatment - MTM1 reduction have a deleterious effect on a long term?

- In this study, DNM2 was downregulated using ASO (and not MTM1 as written in the question). If the question was about DNM2 downregulation long term effect, in the lab we studied this effect in either WT (*Dnm2*^{+/-}) or *Mtm1*KO (*Mtm1*KO *Dnm2*^{+/-}) upon 2 years. A complete behavioural and metabolic characterization was done (unpublished data) and no abnormalities have been noted, suggesting that genetic downregulation of DNM2 is safe. However, in this study, *Dnm2* was downregulated using ASO. Before checking the longterm effects of downregulation of DNM2 through ASOs in a follow-up of this study, we need to better refine the best injection protocol (dose, frequency, pharmacokinetic, window and route of injection ...).

3. As MTM1 is expressed in other tissues what happened to the mice injected with the ASO#1; for instance on what regards the blood cells, nerve or even the NMJ?

- To our knowledge, there is no blood phenotype in myotubular myopathy patients or mouse model. Following the referee 1 request, we checked neuromuscular junctions and nerves in mice treated with ASOs. Concerning nerve, mutations of *MTMR2* (a member of the myotubularin family and close homologue of *MTM1*) are implicated in Charcot-Marie tooth neuropathy. To our knowledge no reported data describe a nerve phenotype neither in MTM1 patients nor *Mtm1*KO mice. Analysis of sciatic nerve stained with blue toluidine revealed normal nerve histology of *Mtm1*KO treated with 25mg/kg of ASO#1 at 12 weeks of age (Rf. Supplementary 11c + text line 185-186). Concerning NMJ, it has been shown that *Mtm1*KO mice exhibit a NMJ defect with an enlarged NMJ area (Dowling et al. 2012). In our cohort, we found that *Mtm1*KO treated with ASO ctrl presented a large NMJ areas and this defect was not restored by downregulating DNM2 with ASO#1 (Rf. supplementary figure 11a, b and text line 180- 189).

4. What happened to the WT mice injected with the ASO#1? There is no description of it?

- This was a very important control group that we didn’t include in the first submission version (except for lifespan and body weight in figure 3a and 3b respectively) because we didn’t see any difference between WT injected with ASO ctrl and those injected with ASO#1. Upon request, all the data of WT injected with 25mg/kg of ASO#1 were added to:

-Figure 3c-i

- Supplementary figure 2a- c
- Supplementary figure 3a, b
- Supplementary figure 4a-f
- Supplementary figure 5a-e
- Supplementary figure 8a, b
- Text line 159-160

5. It should be explained why the authors chose one ASO over the others for the systemic studies.

- We selected ASO#1 for systemic injections because this candidate gave the best results when administrated locally by intramuscular injection. Indeed, the three candidates ASO#1, #2 and #3 restored muscle histology, weight and force. However, only ASO#1 was able to increase muscle force without any difference compared to WT groups (Rf. figure 2d). We concluded that ASO#1 was the best candidate and was chosen for systemic injection. The reason of choosing ASO#1 was previously briefly stated in manuscript (Rf. line 77-78). An additional sentence was integrated to the text (Rf. line 88-90) to better explain this point: “ASO#1 was selected due to the strong rescue effects observed when administrated locally into muscle: namely it had the best dose response (Fig. 1b,c) and increased muscle force to normal levels (Fig. 2d).”

Reviewer #2 (Remarks to the Author):

The work presented by Tasfaout et al. extends a previous study by Dr. Laporte’s laboratory showing that dynamin-2 is upregulated in myotubular myopathy and that reducing dynamin-2 expression is sufficient to improve XLCNM. The current work builds on this idea and now utilizes anti-sense oligonucleotide technology to reduce dynamin-2 expression and prevent myotubular myopathy in a mouse model. This manuscript evaluates a large number of physical and histological parameters and shows that a reduction in dynamin-2, through ASO delivery, is sufficient to extend the lifespan of *Mtm1KO* mice and significantly improve muscle strength. This manuscript is of high technical quality, well constructed, and should interest a large number of readers.

1) The DSS score includes breathing rhythm in the scoring system. What parameters were used to measure respiration? Does ASO treatment improve these respiratory parameters in *Mtm1KO* mice? Additionally, in diaphragm muscle, is there a similar improvement in histology, T-tubule morphology, and force with systemic ASO treatment?

- We apologize for not giving the detailed description of our DSS scoring system. A full description of how DDS was measure was added to the manuscript (Rf . DSS scoring in Material method). DSS score was based on clinical observations and breathing alteration scores was qualitative. *Mtm1KO* treated with ASO ctrl presented an accelerated breathing rhythm with shortness of breath at week 6-7 of age, noticeable by visual inspection and linked to a specific noise. However, ASO#1 treatment restores this phenotype in *Mtm1KO* mice.

- Following the reviewer’s suggestion, WT and *Mtm1KO* diaphragm were analysed. Transversal section histology and ultrastructure analysis by electron microscopy showed that *Mtm1KO* mice treated with ASO ctrl showed a very thin diaphragm composed by hypotrophic fibers with disorganized sarcomere and almost no visible triad, while *Mtm1KO* mice treated with ASO#1 had a normal histology and width, similar to WT, with almost a normal ultrastructure and triads (Rf. manuscript text line 142-153, Supplementary Fig. 7a, b). In addition, we performed immunohistofluorescence experiments (not requested by the reviewer#2 for diaphragm) on RYR1 and DHPR, two proteins located in the triad. Both were mislocalized in *Mtm1KO* fibers when treated with ASO Ctrl. These two proteins were present in the typical transversal double row localization in WT and *Mtm1KO* treated with ASO#1 (Rf. Supplementary Fig. 7a, manuscript text line 148-151) supporting that diaphragm muscle and myofibers structure were restored. Unfortunately, we are not equipped to measure diaphragm muscle force. We measured diaphragm muscle force in a previous study in collaboration with other groups and due to logistic issues (building and sending mice cohorts, availability of material) and the timeline suggested by the editor, we apologize for not being able to answer this request. We hope our additional histology, immunostaining and ultrastructure data are sufficient to sustain a rescue of diaphragm phenotypes upon treatment.

2) Caveolin-3 aggregates in myotubular myopathy. Does ASO treatment ameliorate CAV3 aggregation in

myofibers? RyR and DHPR protein expression are decreased with disease progression and are also mislocalized. Does ASO injection rescue this phenotype?

- Caveolin-3, RYR1 and DHPR localisation was assessed by immunofluorescence on Tibialis anterior transversal or/and longitudinal sections (Rf. supplementary Figure6). While Caveolin-3 was present at the sarcolemma in WT mice at 7 or 12 weeks, *Mtm1KO* treated with ASO ctrl presented many fibers with internal Caveolin-3 aggregates. This phenotype was reduced in *Mtm1KO* when treated with ASO#1 at 7 and 12 weeks. On longitudinal sections, *Mtm1KO* treated with ASO ctrl presented RYR1 aggregates within the fiber; however *Mtm1KO* treated with ASO#1 showed a normal striated staining of RYR1 and DHPR (Rf. supplementary figure 6). In addition, RYR1 and DHPR protein expression was measured by western blot in TA muscle lysate. Both RYR1 and DHPR level were slightly decreased in *Mtm1KO* mice when treated with ASO ctrl or ASO #1 but this was not statistically significant in these cohort at that age ($n=4$ per each group) (supplementary figure 6b, c, d). Overall, the defects seen for Caveolin-3, RYR1 and DHPR in *Mtm1KO* mice are corrected upon treatment, and these points were added in the text (Rf. manuscript text line 132-138).

3) Does systemic ASO#1 treatment, injected after disease onset (Figure 4), also improve the triad/sarcomere ratio and T-tubule morphology?

- Muscle ultrastructure was analysed by electronic microscopy. Seven weeks after the beginning of ASO#1 treatment, *Mtm1KO* mice presented improvement in the myofibers ultrastructure with Z-line alignment and normal T-tubule morphology. The quantification of triad/sarcomere ratio showed that treated *Mtm1KO* muscles presented an increase in triad number. Note that at that age, *Mtm1KO* mice treated with ASO ctrl are dead; these novel data could be compared to 7 week old *Mtm1KO* mice treated with ASO ctrl (Fig. 3e and supplementary Fig. 3b). These novel data were included in figure 4 (Fig. 4h, k) and in the text (Rf. manuscript text line 211-212).

4) Could the authors comment on any off target effects of DNM2 reduction as DNM2 is ubiquitously expressed or if they have experienced detrimental effects with decreasing DNM2 beyond a certain threshold?

- ASO are known to accumulate in liver and kidney and induce hepatotoxicity and nephrotoxicity when administrated in high doses. We measured liver and kidney function markers in blood samples (ASAT, ALAT, creatinine and urea). These markers were normal in both WT and *Mtm1KO* that were treated with ASO ctrl or ASO#1 despite a strong DNM2 decrease especially in the liver (supplementary Figure10). We noted a few hepatocyte with enlarged cytoplasm and condensed nuclear material only in WT mice treated with 25mg/kg of ASO#1 but not in *Mtm1KO* mice (Rf. manuscript text line 164-171). This could be correlated to the increase of ASO concentration in these samples, as measured in supplementary Fig. 10c. Except these finding, liver and kidney histology were normal in WT and *Mtm1KO* mice treated with 25mg/kg ASO ctrl or ASO#1.

5) What was the level of DNM2 reduction in the severe, dead mice?

- We didn't analyze DNM2 level in *Mtm1KO* mice *post mortem* because we believe that protein expression can vary in dead mice through protease effects and other cellular process like autophagy.

6) Figure 3a and 3b show data from WT+25mg/kgASO#1. The data from this cohort is not included in 3c-i. Were there any significant effects noted in this cohort of mice for any of the parameters analyzed in 3c-i?

- This was a very important control group that we didn't include in the first submission version (except for lifespan and body weight in figure 3a and 3b respectively) because we didn't see any difference between WT injected with ASO ctrl and those injected with ASO#1. Upon request, all the data of WT injected with 25mg/kg of ASO#1 were added to:

- Figure 3c-i
- Supplementary figure 2a- c
- Supplementary figure 3a, b
- Supplementary figure 4a-f

- Supplementary figure 5a-e
- Supplementary figure 8a, b
- Text line 159-160

7) In Figure 3, 25mg/kg ASO injections resulted in DNM2 levels that were 50% of wildtype levels. In Figure 4, this dosage, starting two weeks later, resulted in DNM2 levels equivalent to wildtype. What is the timing of DNM2 reduction with ASO injection? Please explain this difference in DNM2 expression.

- Indeed, at age of 12 weeks, DNM2 knockdown in *Mtm1*KO varies when ASO#1 is administrated at week 3 for disease blockade and prevention (figure 3h) or week5 for disease reversion (figure 4l). This difference could be explained by the co-association of two factors: date at the beginning of the treatment and disease progression. In fact, we hypothesize that at week3 the *Mtm1*KO mice are slightly affected and DNM2 level might be similar to WT level, whereas at week5 *Mtm1*KO mice are severely affected and DNM2 level is more strongly upregulated. Thus, the sooner ASO#1 is administrated, the better is the DNM2 knockdown. We also noted in the discussion that, “while DNM2 protein level is increased in the disease state, disease reversion correlated with DNM2 level normalization (Fig. 4l), suggesting it is not necessary to decrease DNM2 below normal level to reach a benefit” (Rf. Text line 224-227).

Concerning the timing of DNM2 reduction after ASO administration, two cohorts of wild-type mice were treated by single injection of ASO ctrl or ASO#1 at week3. Then, they have been sacrificed one or two weeks later. We found that after one week DNM2 was reduced to around 80% while it was down regulated to 50% after two weeks (Rf. supplementary figure 9, text line 160-162). It seems that in WT mice, reduction of DNM2 to 50% is achievable 2 weeks after one ASO#1 injection. However, as *Mtm1*KO mice present an upregulation of DNM2 through disease progression, it would be difficult to extrapolate these findings. While similar tests might be done on *Mtm1*KO mice with a single shot of ASO#1, the limit that we foresee is the short lifespan of this mouse model: indeed such study would be possible only if a single injection significantly extend the lifespan of these animals.

Reviewer #3 (Remarks to the Author):

Myotubularin is a lipid phosphatase centrally involved in muscle membrane homeostasis, as shown also by the fact that mutations in the human MTM1 gene cause myotubular myopathy, a severe recessive X-linked disease of the centronuclear myopathy group with early onset and premature death. CNM can also be caused in dominant form by mutations in dynamin-2 (Dnm2), a GTPase that is intriguingly involved in vesicular traffic or microtubule dynamics.

This study builds on earlier observations that the double knockout of myotubularin and dynamin-1 (Mtm1, Dnm2) rescues the myopathic phenotype causes by Mtm1 deficiency alone. The authors, by translating these findings to an antisense oligonucleotide-based down-regulation of Dnm2, significantly show that this Dnm2 reduction can prevent and even revert myotubular myopathy symptoms in the Mtm1 mouse model in a dose-dependent way, and thus open the door to a potential therapeutic approach for this rare but devastating myopathy.

These results are very impressive- all quantitative parameters show a dramatic improvement of morphology (including correction of the central feature of myotubular myopathy, the centronucleation) and functional performance, as well as a significant extension of life-span in those mice that survive beyond the first week of treatment after age 5 weeks.

Apart from this translational angle, the work also firmly supports the mechanistic interplay between Dnm2 and Mtm1 in basic muscle biology and is therefore also of fundamental interest.

This is overall a well-written and illustrated, experimentally and conceptually excellent paper that offers both intriguing biological and pathomechanistic insight. There are some questions that the authors should address, however.

Comments:

1. Please state the gender of the animals used in the experiments!

- We apologize forgetting to precise that in this study all the experiments were performed on male 129SvPAS mice. This information on the mice gender was added to the manuscript text in line 250.

2. The GAPDH control blot in Figure 3h seems identical to that in Supplemental Figure 6, yet the dosage regimes and genotypes would be different.

- We would like to thank the reviewer for this observation and we sincerely apologize for this unintentional mistake. Indeed, the same GAPDH blot was copied in figure 3h and supplementary figure 6 due to a mistake in selecting the image files. The correct GAPDH loading control of the previous supplementary figure 6 was corrected (Rf. Supplementary Fig 8). We would like to pointout that quantification graphs of the first submission version were correct and were done with the appropriate GAPDH bands.

3. Are the differences in survival in Figure 4b statistically significant?

- To test if survival difference is statistically significant, Gehan-Breslow-Wilcoxon test has been performed. Survival rate of *Mtm1*KO treated with 25mg/kg of ASO#1 is not significant when compared to WT ASO ctrl ($p=0.06$) nor *Mtm1*KO ctrl ($p=0.45$). However, *Mtm1*KO treated with ASO ctrl had a significant reduction of survival when compared to WT treated with ASO ctrl ($p=0.0017$)

4. Specific force in WT control mice in Figure 3d is somewhat lower than in 4g- please explain the difference (age?)

- Indeed, the difference in specific muscle force (the ratio absolute TA muscle force/TA muscle weight) of WT treated with ASO ctrl in Fig. 3d (20.43 ± 3.81) is slightly lower compared to WT (26.35 ± 2.86) in Fig.4g. This is most probably due to age of the mice. Figure 3d showed muscle force at 7weeks while figure 4 muscle force was measured at week 12.

5. What is the explanation why in Figure 4k there is no decrease in Dnm2 below the normal levels compared to WT?

- Indeed, at age of 12weeks, DNM2 knockdown in *Mtm1*KO varies when ASO#1 is administrated at week3 for disease blockade and prevention (figure 3h) or week5 for disease reversion (figure 4l). This difference could be explained by the co-association of two factors: date at the beginning of the treatment and disease progression. In fact, we hypothesize that at week3 the *Mtm1*KO mice are slightly affected and DNM2 level might be similar to WT level, whereas at week5 *Mtm1*KO mice are severely affected and DNM2 level is more strongly upregulated. Thus, the sooner ASO#1 is administrated, the better is the Dnm2 knockdown. From our data, starting an ASO treatment at week5 will normalize the DNM2 level at week12. We also noted in the discussion that, "while DNM2 protein level is increased in the disease state, disease reversion correlated with DNM2 level normalization (Fig. 4l), suggesting it is not necessary to decrease DNM2 below normal level to reach a benefit" (Rf. Text line 224-227).

6. It would be good to indicate the molecular weights for the bands in the Western blots of the figures.

- Following this request, the molecular weight of all bands was added in:

- Figure 1c
- Figure 2a
- Figure 3h
- Figure 4l
- Supplementary Fig. 6b
- Supplementary Fig. 8 b
- Supplementary Fig. 9 a
- Supplementary Fig. 10d

Reviewer #4 (Remarks to the Author):

In this paper, Tasfaout et al. examined the effects of antisense-oligonucleotide-mediated knockdown of Dnm2 in a mouse model of myotubular myopathy. The author found that the novel oligonucleotides significantly reduce the expression of Dnm2, accompanied by the ameliorated pathology in mutant mice. The finding is interesting

and promising. It will provide meaningful information that is of use in designing therapeutic strategy. However, there are many flaws and points that should be addressed to strengthen author's claim.

Main Points

1. No toxicology data is included. Blood biochemistry (e.g. BUN, AST, ALT etc.) and histology of kidney and liver should be included. Some oligonucleotides (e.g. 2'MOE) accumulate in kidney, and this possibility needs to be examined.

- As requested, we measured liver and kidney function markers in blood samples (ASAT, ALAT, creatinine and urea). These markers were normal in both WT and *Mtm1*KO that were treated with ASO ctrl or ASO#1 despite a strong DNMT2 decrease especially in the liver (novel supplementary Figure 10). We noted a few hepatocyte with enlarged cytoplasm and condensed nuclear material only in WT treated with 25mg/kg of ASO#1 but not in *Mtm1*KO mice. This could be correlated to the increase of ASO concentration in these samples, as measured in supplementary Fig. 10c. Except these finding, liver and kidney histology were normal in WT and *Mtm1*KO mice treated with 25mg/kg ASO ctrl or ASO#1 (Rf. manuscript text line 166-171).

2. Line 95- The authors' statement that injections with 6.25...25 mg/kg doses improved muscle weight is not supported by the data. The figure shows no statistical significance except for the 25 mg/kg group (fig 3c).

- We would like to thank the reviewer for this observation and we sincerely apologize for this unintentional mistake. Indeed, injection of 6.25 mg/kg of ASO#1 did not improve neither TA nor gast. muscle weight and this was corrected in the text (Rf. manuscript line 107).

3. Line 142- The authors' claim that the mice could not be rescued by a single injection contradicts with the statement in line 145 that after only one injection mice improved quickly. From the figure, it appears multiple injections are needed.

- We would like to thank again the reviewer for this observation. We meant that the *Mtm1*KO mice dying in the week following the first injection could not be rescued after one injection due to phenotype severity; however, rescued *Mtm1*KO presented a phenotype stabilization after one injection and the disease was rescued after the second injection. We have now better clarified this point in the manuscript: "After only one injection, surviving *Mtm1*KO mice presented a stabilization of their phenotype, and were rescued after a two weeks treatment" (Rf. Manuscript text line 204-205).

4. In several figures, the t-test is used inappropriately (e.g. fig 1, 2, suppl. fig 3). ANOVA should be used to compare multiple groups.

- We apologize for this inappropriate test; *t*-test was replaced by ANOVA to allow the multiple groups comparison in previous analyses and was used in novel figures :

- Figure 1b, c
- Figure 2a, c, d, f, g
- Supplementary figure 2a-c
- Supplementary figure 3b
- Supplementary figure 4b, f
- Supplementary figure 5d, e
- Supplementary figure 6
- Supplementary figure 7b
- Supplementary figure 8a, b
- Supplementary figure 10b, c, e
- Supplementary figure 11b

5. In addition, in several figures (e.g. fig 1), the comparison is made against non-treated groups, however, they should be compared against ctrl ASO.

-We have modified this point and data are now comparing to ctrl ASO treated group (Rf. Figure 1b, c).

6. The authors examined the effects only up to 12 weeks. How long does the effect persist?

- This is indeed an interesting question. We stopped the study at 12 weeks because we observed an obvious phenotypic amelioration in addition to a doubling lifespan of *Mtm*1KO mice when they were treated with 25mg/kg of ASO#1. We plan to address this point in a follow-up study, as it requires first to better refine the best injection protocol (dose, frequency, pharmacokinetic, window and route of injection ...) and producing an important quantity of ASO.

7. Line 129- : The authors claim that a significantly higher distribution was observed with both control ASO and ASO #1 in muscles from the mutant mice compared to WT, however, this is not supported by the data. The figure does not show statistics. In addition, authors do not explain the mechanisms underlying the differences.

- The statistical tests have been done (Rf, Figure 3i, Supplementary 8a).

- The mechanism of uptake of ASO in muscle is not understood. Moreover, the function of MTM1 and DNM2 in muscle was barely investigated to date, and never in relation to ASO kinetic. We can hypothesize that the absence of MTM1 impacts on the uptake or the degradation of ASO.

8. In figure 1 and 2, the number of experiments is not shown.

- The number of experiment has been added to the manuscript (Rf. line 390, 397, 398, 399, 400, 402 and 403)

9. In Figure 3h, there are two bands that are unexplained.

- In unpublished data from the lab, we verified the two bands are DNM2 with different antibodies and through the fact that they are both decreased upon DNM2 knockdown or knockout (a *Dnm2*^{+/-} is available).

Indeed, in western blot from adult mouse organ lysates DNM2 appears in two bands (figure 3h, 4l Supplementary Figs 8, 9 and 10) while in myoblast cell lysates, we were able to detect only the upper band (figure 1c). This suggests that the second band could be either an isoform (result of alternative splicing), a post-transcriptional modification or a degradation product. A sentence was added to legend of Figure 2 (Rf. text line 396).

10. The histology data (figure 3e) are too small and hard to evaluate.

- The pictures were magnified for a better evaluation of the muscle histology (Rf Fig. 2e, Fig. 3e, Fig. 4h, Supplementary Fig 5c).

11. Figure 4 J and K: The authors claim that no significant differences were observed between rescued mice and WT but no detail about the statistical analysis is given.

- Indeed, in figure 4 we did not find any statistical difference in fiber size ($p= 0.13$) or DNM2/GAPDH ratio ($p=0.55$). To avoid misinterpretations, NS symbol (no statistical difference) was added above the columns and explained in the legend (Rf Figure 4j, 4l)

Minor Points

1. Apparently, the authors employ novel antisense oligos but the manuscript is hard to follow without the structure of the oligonucleotide.

- A representation of the chemical structure of the phosphorothioate constrained ethyl ASO was added to the manuscript (Rf. Supplementary table1)

2. There should be a space between the numerical value and unit symbol

- This has been corrected throughout the text.

REVIEWERS' COMMENTS:

Reviewer #2 (Remarks to the Author):

The revisions have significantly improved the manuscript and satisfies this Reviewer.

Reviewer #4 (Remarks to the Author):

In this revised manuscript, the authors have done well to answer the reviewers' comments— with additional experiments and changes to the main text—, however, there are still some issues to be resolved and further additions/edits that can be done to improve the strength of the manuscript. Here are my comments:

1. It would be good to include a description of the antisense oligonucleotide used (i.e., the cET chemistry) in the introduction. Since it is relatively new, it may not be familiar to most readers. Also, what is the reason for using this chemistry? What are its advantages and disadvantages to, say, using locked nucleic acids instead? Is it safer or more efficient? These have to be mentioned.
2. In Figure 1c, why does it seem as if the 0.25 μ M dose were better than the 1.00 μ M dose?
3. Figure 2c was not referenced in the main text and so its purpose was not highlighted.
4. Can it be explained why the result in Fig. 3h (last two lanes with 25 mg/kg ASO #1 in Mtm1 KO) is different from that in Fig. 4l (rescued Mtm1 KO lanes)?
5. Some typographical errors:
 - a. Line 16: supposed to be “myotubularin” instead of “myotubular”?
 - b. Lines 103, 108, 349 have wrongly-spaced words.
 - d. Line 205: change “two weeks” to “two-week”
 - e. Line 313: misspelled “properties”
6. The discussion could be further enriched to explore more aspects of the study. The following topics are suggested:
 - a. What are the limitations of the study? The discussion seems to be too one-sided, with no discussion of possible drawbacks of the treatment, possible factors that have been missed in the experiments, sub-optimal or unwanted results, and other such details.
 - b. Will it be a good idea to check for off-target effects at the gene level through methods like RNA sequencing?
 - c. Since the authors have already touched on how their study sheds some light on the possible interaction between Mtm1 and Dnm2, it may be helpful to discuss the normal roles of these two genes and their respective proteins (using previous literature) and how the authors think their treatment is working with respect to how it affects cellular structures/processes besides the fact that it decreases Dnm2 expression. Alternatively, information on Mtm1 and Dnm2 can be presented in the introduction.

Answer to questions of Referee #4 :

Reviewer #4 (Remarks to the Author):

In this revised manuscript, the authors have done well to answer the reviewers' comments—with additional experiments and changes to the main text—, however, there are still some issues to be resolved and further additions/edits that can be done to improve the strength of the manuscript. Here are my comments:

1. It would be good to include a description of the antisense oligonucleotide used (i.e., the cET chemistry) in the introduction. Since it is relatively new, it may not be familiar to most readers. Also, what is the reason for using this chemistry? What are its advantages and disadvantages to, say, using locked nucleic acids instead? Is it safer or more efficient? These have to be mentioned.

➤ Upon this request, an additional paragraph was added to the main text (line 53-56) to describe the reasons of using the ethyl constrained chemistry.

2. In Figure 1c, why does it seem as if the 0.25 uM dose were better than the 1.00 uM dose?

➤ DNM2 protein level in cells treated with 0.25µM or 1.00µM was not statistically different.

3. Figure 2c was not referenced in the main text and so its purpose was not highlighted.

➤ Figure 2c represents ASOs effect on TA muscle weight upon local injection of these drugs. This figure was referenced in the main text (previous version line 77, new revised version line 83-86).

4. Can it be explained why the result in Fig. 3h (last two lanes with 25 mg/kg ASO #1 in *Mtm1* KO) is different from that in Fig. 4l (rescued *Mtm1* KO lanes)?

➤ This question was already asked by **Referee #2 question 7** and **Refree#3 question 5**. The explanation is that DNM2 knockdown in *Mtm1*KO varies when ASO#1 is administrated at week 3 for disease blockade and prevention (previous version figure 3h, new revised version figure 7a) or week5 for disease reversion (previous version figure 4l, new revised version figure 9h). This difference could be explained by the co-association of two factors: date at the beginning of the treatment and disease progression. In fact, we hypothesize that at week3 the *Mtm1*KO mice are slightly affected and DNM2 level might be similar to WT level, whereas at week5 *Mtm1*KO mice are severely affected and DNM2 level is more strongly upregulated. Thus, the sooner ASO#1 is administrated, the better is the DNM2 knockdown. We also noted in the discussion that, “while DNM2 protein level is increased in the disease state, disease prevention with 12.5 mg/kg (**Fig. 7b**) and disease reversion with 25 mg/kg (**Fig. 9h**) both correlated with DNM2 level normalization, suggesting it is not necessary to decrease DNM2 below normal level to reach a benefit while minimizing the potential risk of DNM2 target reduction.” (Rf. Text line 244-248).

5. Some typographical errors:

a. Line 16: supposed to be “myotubularin” instead of “myotubular”?

b. Lines 103, 108, 349 have wrongly-spaced words.

d. Line 205: change “two weeks” to “two-week”

e. Line 313: misspelled “properties”

- We thank the referee for these remarks, corrections have been made.

6. The discussion could be further enriched to explore more aspects of the study. The following topics are suggested:

- a. What are the limitations of the study? The discussion seems to be too one-sided, with no discussion of possible drawbacks of the treatment, possible factors that have been missed in the experiments, sub-optimal or unwanted results, and other such details.
- b. Will it be a good idea to check for off-target effects at the gene level through methods like RNA sequencing?
- c. Since the authors have already touched on how their study sheds some light on the possible interaction between Mtm1 and Dnm2, it may be helpful to discuss the normal roles of these two genes and their respective proteins (using previous literature) and how the authors think their treatment is working with respect to how it affects cellular structures/processes besides the fact that it decreases Dnm2 expression. Alternatively, information on Mtm1 and Dnm2 can be presented in the introduction.

- As requested, the discussion was expended and the following points were discussed:
- Long-term effect of reducing Dnm2 trough ASOs (line: 229-236).
 - Study limits (reversion protocol) (line: 238-248).
 - MTM1 and DNM2 functional interaction (line 249-256).
 - Some perspectives for the human candidate and clinical trials (line 259-263).